# CliBench: A Multifaceted and Multigranular Evaluation of Large Language Models for Clinical Decision Making

## Abstract

The integration of Artificial Intelligence (AI), especially Large Language Models (LLMs), into the clinical diagnosis process offers significant potential to improve the efficiency and accessibility of medical care. While LLMs have shown some promise in the medical domain, their application in clinical diagnosis remains underexplored, especially in real-world clinical practice, where highly sophisticated, patient-specific decisions need to be made. Current evaluations of LLMs in this field are often narrow in scope, focusing on specific diseases or specialties and employing simplified diagnostic tasks. To bridge this gap, we introduce CliBench, a novel benchmark developed from the MIMIC IV dataset, offering a comprehensive and realistic assessment of LLMs' capabilities in clinical diagnosis. This benchmark not only covers diagnoses from a diverse range of medical cases across various specialties but also incorporates tasks of clinical significance: treatment procedure identification, lab test ordering and medication prescriptions. Supported by structured output ontologies, CliBench enables a precise and multi-granular evaluation, offering an in-depth understanding of LLM's capability on diverse clinical tasks of desired granularity. We conduct a zero-shot evaluation of leading LLMs to assess their proficiency in clinical decision-making. Our preliminary results shed light on the potential and limitations of current LLMs in clinical settings, providing valuable insights for future advancements in LLM-powered healthcare.

## 1 Introduction

An accurate diagnosis is pivotal for delivering effective medical care, involving the identification of diseases and therapeutic management based on a comprehensive analysis of patient demographics, symptoms, medical history, and diagnostic test results. It is a complex cognitive process that requires extensive medical knowledge, reasoning, and experience. In the era of digital healthcare, building AI systems that can automate or assist clinicians in this process with high accuracy has profound implications for reducing healthcare costs and enhancing the accessibility of medical expertise.

Recent advancements in Large Language Models (LLMs) (OpenAI, 2024; Anil et al., 2023) have opened up new possibilities. They have demonstrated remarkable capabilities in text understanding, reasoning, and generating proper responses in various domains (Ma et al., 2024b), including the medical field (Singhal et al., 2023a;b; Ma et al., 2023; Xu et al., 2023). Specifically, LLMs have been thoroughly evaluated and shown to excel in medical licensing examinations (Nori et al., 2023b; Gilson et al., 2023) and medical knowledge QA benchmarks (Singhal et al., 2023b; Nori et al., 2023a; Toma et al., 2023). However, the application of LLMs in clinical diagnosis represents a more nuanced and realistic challenge, requiring not only the understanding of medical knowledge but also the ability to make complex clinical decisions based on real-life patient-specific scenarios (McDuff et al., 2023; Tu et al., 2024).

Nevertheless, prior evaluations of LLMs about clinical diagnosis capabilities reveal significant constraints. Most research has either concentrated on diagnosing singular diseases (Kwon et al., 2023; Hager et al., 2024) or restricted its focus to certain medical specialties (Allahqoli et al., 2023; Krusche et al., 2024), lacking the breadth necessary for general clinical practice. With the small coverage of disease types, such studies often employ simplified task formats, requiring the model to

choose a diagnosis from binary or multiple choices of diagnostic candidates, reducing the intricacy of real-world clinical decision-making. Moreover, most studies predominantly target at evaluating the performance of LLM in diagnostic predictions (Takita et al., 2024; Ma et al., 2024c), while omitting the other critical clinical decisions for thorough patient care, such as the ordering of various lab tests and the arrangement of all follow-up treatment procedures. Table 1 shows a comprehensive list of current studies of LLMs in clinical diagnosis and their limitations as mentioned earlier.

To address these gaps, we introduce a novel benchmark, CLIBENCH, aimed at a more accurate and inclusive assessment of LLMs' capabilities within the realm of clinical diagnosis. Meticulously curated from the MIMIC IV dataset (Johnson et al., 2023), our benchmark spans a broad spectrum of cases across various specialties, enriched by a connection to a structured expert-curated diagnosis ontology, the ICD-10-CM coding (icd, 2023a) for precise and hierarchical evaluation. Moreover, our benchmark extends beyond mere diagnostic capabilities, challenging LLMs to also recommend treatment procedures, formulate lab test orders and prescribe medications adhered to the ICD-10-PCS (icd, 2023b) ontology, LOINC coding system and ATC classification system. Furthermore, we provide a dataset construction pipeline that is designed to support not only evaluation but also the generation of training data, facilitating ongoing training-based improvements in model performance.

We conduct experiments using zero-shot configurations of prominent open-sourced and close-sourced LLMs, covering both general-purpose and medical-domain models, to assess their clinical decision-making performance. Our evaluation is hierarchical, reflecting the complexity and multi-faceted nature of clinical diagnosis. Preliminary results from these experiments highlight the strengths and weaknesses of current LLMs in making clinical decisions, offering insights into areas for further research and development.

## 2 RELATED WORKS

**Clinical capabilities benchmarks for LLMs.** One of the primary benchmarks for evaluating LLMs in clinical settings is their mastery of medical knowledge, particularly in the task format of question-answering. The widely-used MultiMedQA (Singhal et al., 2022) benchmark suite comprises seven medical QA benchmark datasets: MedQA (Jin et al., 2021) and MedMCQA (Pal et al., 2022) consists of multiple choice questions from medical exams, PubMedQA (Jin et al., 2019) sources research questions from PubMed literature, subsets of MMLU (Hendrycks et al., 2021) covers some medically relevant topics, and LiveQA (Abacha et al.), MedicationQA (Abacha et al., 2019), and HealthSearchQA (Singhal et al., 2022) was curated from commonly asked consumer questions. These benchmarks from medical exams, research papers, and common questions, though effective in assessing the general medical knowledge of LLMs, significantly differ from the daily complexities and importance of clinical decision-making faced by clinicians in real-life cases.

**Evaluation of LLMs on clinical diagnosis.** A growing body of studies has explored the capabilities of LLMs in clinical diagnosis, each with diverse target scopes and task configurations. In Table 1, we comprehensively compare the current evaluation studies of LLM's diagnostic performance, listing the heterogeneity and limitations across studies.

The literature predominantly centers on evaluations with the **Focus of Medical Specialties** on either a *single disease* such as Alzheimer's Disease (Kwon et al., 2023), Appendicitis, Cholecystitis, Diverticulitis, and Pancreatitis (Hager et al., 2024), or a *single specialty* such as Ophthalmology (Lyons et al., 2023; Hu et al., 2023; Knebel et al., 2023; Rojas-Carabali et al., 2023; Madadi et al., 2023; Sorin et al., 2023), Neurology (Horiuchi et al., 2024; Galetta & Meltzer, 2023; Koga et al., 2023), and Dermatology (Stoneham et al., 2023; Rundle et al., 2024; Ravipati et al., 2023). A few works conduct their evaluation containing cases from *multiple specialties*, but the **Dataset Size (# Test Cases)** of these works is often less than 100, hardly reaching 500, resulting in *limited* coverage of disease types. Considering these two key features of diagnostic target scope, the current studies often narrow the disease spectrum and fail to mirror the wide breadth needed for general clinical practice.

With the small range of targeted disease types, these studies often simplify the **Diagnostic Output Scope** to a *binary* decision of whether a certain disease should be diagnosed or not (Mori et al., 2023; Hager et al., 2024), or to a *multiple-choice* question with 3-5 curated negative choices (Eriksen et al., 2023; Han et al., 2023). These task settings largely reduce the complexity of the real-life clinical decision-making process. Some studies offer a more realistic setting by generating *free text* responses, but they mainly require extensive manual checking at the expert level to evaluate the

Table 1: Comparison of CLIBENCH with other LLM evaluation studies on clinical diagnosis.

| Studies | Focus of Medical Specialties | # Test Cases | Diagnostic Output Scope | Clinical Decision Targets | Availability of Training Data |
|---|---|---|---|---|---|
| Mori et al. (2023) | Disease-Specific | 151 | Binary | Diagnoses | ✗ |
| Kwon et al. (2023) | Disease-Specific | 1187 | Binary | Diagnoses | ✓ |
| Hager et al. (2024) | Disease-Specific | 2400 | Binary | Diagnoses, Lab Tests, Procedures | ✗ |
| Kiyohara et al. (2023) | Specialty-Specific | 66 | Multiple-Choice | Diagnoses | ✗ |
| Abi-Rafeh et al. (2023); Allahqoli et al. (2023); Bushuven et al. (2023); Daher et al. (2023); Gebrael et al. (2023); Knebel et al. (2023); Lyons et al. (2023) | Specialty-Specific | ≤ 100 | Free Text | Diagnoses, Procedures | ✗ |
| Berg et al. (2024); Brin et al. (2023); Chee et al. (2023); Delsoz et al. (2023a); Delsoz et al. (2023b); Fraser et al. (2023); Horiuchi et al. (2023a); Horiuchi et al. (2023b); Horiuchi et al. (2024); Hu et al. (2023); Koga et al. (2023); Krusche et al. (2024); Madadi et al. (2023); Mitsuyama et al. (2023); Nakaura et al. (2024); Pillai & Pillai (2023); Ravipati et al. (2023); Rojas-Carabali et al. (2023); Rundle et al. (2024); Sorin et al. (2023); Stoneham et al. (2023); Suthar et al. (2023); Tenner et al. (2024); Ueda et al. (2023a); Xv et al. (2023) | Specialty-Specific | ≤ 500 | Free Text | Diagnoses | ✗ |
| Andrade-Castellanos et al. (2023); Benoit (2023); Hirosawa et al. (2023b); Hirosawa et al. (2023a); Ito et al. (2023); Kanjee et al. (2023); Levine et al. (2023); Mykhalko et al. (2023); Reese et al. (2023); Schubert et al. (2023); Shea et al. (2023); Ueda et al. (2023b) | Multi-Specialties (Limited) | ≤ 100 | Free Text | Diagnoses | ✗ |
| Tu et al. (2024) | Multi-Specialties (Limited) | 149 | Free Text | Diagnoses, Procedures | ✗ |
| Eriksen et al. (2023); Han et al. (2023); Ueda et al. (2023b) | Multi-Specialties (Limited) | ≤ 500 | Multiple-Choice | Diagnoses | ✗ |
| **CLIBENCH** | **Multi-Specialties (Comprehensive)** | **1000+** | **Large Expert Ontology** | **Diagnoses, Procedures, Lab Tests, Prescriptions** | ✓ |

response (Rojas-Carabali et al., 2023), lacking structured and efficient evaluation support. Moreover, the **Clinical Decision Targets** are frequently limited to generating *diagnoses*, neglecting other vital clinical decisions that are also essential for comprehensive care, such as recommending appropriate treatment *procedures*, arranging necessary *lab tests* and issuing medication *prescriptions*. Finally, the lack of broad **Availability of Training Data** in these studies impedes the development of models that can generalize well across diverse medical conditions and patient demographics. To bridge these gaps, there is a pressing need for benchmarks that offer a wide-ranging, realistic, effective, and comprehensive evaluation of LLMs in clinical diagnosis.

## 3   CLIBENCH: MULTIFACETED BENCHMARK FOR CLINICAL DECISIONS

We introduce CLIBENCH, a multi-specialty clinical decision evaluation benchmark covering diagnosis, procedure, lab test order, and prescription on real clinical cases (Section 3.1). We perform cross-dataset cross-table data extraction with NLP pipeline and human verification (Section 3.2), diversity-assured evaluation set sampling (Section 3.3), task-specific prompt construction (Section 3.4), flexible natural language output to label matching (Section 3.5) while scoring models under multi-granular settings of various difficulties (Section 3.6).

### 3.1   REALISTIC CLINICAL DECISION TASKS WITH EXPERT-DEFINED OUTPUT SPACES

We introduce four clinical decision tasks: deciding diagnoses, identifying procedures, ordering lab tests and prescribing medications. To reconstruct clinicians' decision processes, different sets of information are provided as input for each task, and the target output space is also task-specific. We show a table laying out the comparisons and examples in Appendix B.1.

**Task 1: discharge diagnoses.** Diagnosis is defined as the identification of a disease, condition, or injury based on a patient's health evidence. The task aims to provide a set of diagnoses according to the *patient profile*, *medical record at admission*, *lab test results within the admission*, *radiology results within the admission* and *history diagnoses*. Each diagnosis is represented in the International Classification of Diseases, tenth Revision, Clinical Modification (ICD-10-CM) code or equivalent concepts(icd, 2023a), which is a coding system used by healthcare providers to classify all diagnoses for claims processing. The history diagnoses are necessary for completeness because diagnoses made in previous admissions or other service departments might be inherited.

**Task 2: procedures.** Procedures are specific courses of action, to be implemented to intervene in the patient's health status. The task aims to identify the first batch (defined in Appendix B.6) of procedure decisions after the patient is admitted. The input contains *patient profile* and *medical record at admission*. The expected output is a set of ICD-10-Procedure Coding System or equivalent concepts (icd, 2023b). Within a certain admission, procedure decisions, lab test orders and prescriptions can be made at any time, where the later decisions are made while the clinician is aware of outcomes and results of previous procedures or lab tests. It is hard to obtain ground-truth non-initial decisions since the actions can be taken in different temporal orders while only the outcomes of the factual action order are available, which motivates us to predict only the first batch of decisions in terms of time.

**Task 3: lab test orders.** With the same input as procedure decisions, the task aims to produce a set of initial lab items after the patient is admitted to facilitate downstream diagnosis and treatment. Each lab item is a unique Logical Observation Identifiers Names and Codes (LOINC) code.

**Task 4: prescriptions.** Given the same input as the procedure decisions, the prescription task yields a set of initial medications to be prescribed for the patient after being admitted. Each medication is coded in the Anatomical Therapeutic Chemical (ATC) classification system.

## 3.2 DATA PROCESSING AND CLINICAL DATA ELEMENTS EXTRACTION

We extract clinical data elements as the foundational information units by cross-referencing multiple tables of the `hospital` and `note` modules of the MIMIC-IV dataset (Johnson et al., 2023), which contains hospital-wide Electronic Health Records (EHR) from 2008 to 2019 at the Beth Israel Deaconess Medical Center in Boston. First, we obtain **patient profile**, including gender, race, age, insurance category, language and marital status. Second, we obtain the **medical record at discharge**, including major complaint, history of present illness, past medical history, social history, family history, physical exam data, allergies, and current medications at the hospital admission time. These patient status records are represented in natural language in discharge notes. We then induce the **medical record at admission** by removing the sections that describe information only available after the clinical decisions (diagnoses, procedures, lab tests, and prescriptions) are made from the medical record at discharge. Examples of those sections include discharge physical exam data, updated medication lists, and doctor's notes/comments. The induction is done by a rule-based NLP pipeline involving section identification, title extraction and keyword-matching. A manual screening process on 1000 sampled records iteratively updates the pipeline rules for coverage. The correctness of all medical records at admission in the evaluation set is manually verified by a clinical NLP expert. Third, **lab test results within the admission** includes lab test item, result value, units, normal range and interpretation flags. Fourth, we identify **radiology results within the admission** including indications and findings of patient radiology scans. Finally, **history diagnoses** are extracted from discharge diagnoses of the last admission of the same patient and represented in ICD-10-CM codes, comprehended with natural language medical history within the medical record at admission.

For ground-truth labels, we use billing diagnoses in ICD-10-CM codes as an estimation of discharge diagnoses, and we extract procedures, lab tests and prescriptions ordered by a clinician at the first timestamp after admission from structured records in ICD-10-PCS, LOINC and ATC codes at the hospital encounter. All decisions are unordered sets of codes.

## 3.3 BALANCED AND DIVERSE EVALUATION INSTANCES SAMPLING

We sample a representative evaluation set from the extracted admission instances for each task, and the remaining instances can be used for training. There are no overlapped admissions or patients between split sets. We filter out the admission instances where there is no records for the target element (*i.e.* diagnoses, procedures, lab test orders, or prescriptions).

We sample the evaluation set by complying with the following distributional requirements. First, we expect the evaluation set to cover a diverse and broad range of output space. To do so, we perform balanced sampling for different chapters of ICD-10-CM diagnosis codes (*e.g.* E00-E89 for "endocrine, nutritional and metabolic diseases"), top-level categories for ICD-10-PCS procedure codes (*e.g.* "imaging" procedures), third-level categories for LOINC lab items (*e.g.* "drug toxicology laboratory" under LP7790-1), and ATC 1st level (*e.g.* chemicals for the respiratory system) for diagnosis, procedure, lab test order, and prescription tasks, respectively. Second, we require scenarios of various service and care units to be included in the evaluation set. Each admission record is associated with corresponding *service departments*, which cared for the patient during their

hospitalization, and *care units*, the type of unit or ward in which the patient is physically located. There are 21 service departments ranging from cross-specialty departments like "surgical" to more specialty-specific departments like "cardiac surgery", and 37 care units such as "Medical ICU" and "Coronary Care Unit". After sampling by output space distribution, we additionally sample a minimum number of instances for each service or care unit to comprehend if the per-category data count is not sufficient.

### 3.4 PROMPT CONSTRUCTION

After clinical data elements are extracted and cleaned, we create an input prompt incorporating the system profile (*e.g.* "You are a professional clinician in a hospital with expert knowledge in medical and clinical domains."), task instruction (*e.g.* "The task is to make a list of diagnoses for this patient based on the provided information of the patient..." for the diagnosis task), and verbalized patient record indicated in Section 3.1 for each task. We provide complete prompt examples for each task in Appendix B.2. For long prompts, we truncate the patient record segment by cutting the end of each prompt segment following the same ratio (detailed implementation in Appendix B.7).

### 3.5 MATCHING FROM FREE TEXT GENERATION TO MEDICAL CODE CANDIDATE

We design a flexible mechanism to convert LLM output to the modality required by tasks. In clinical operations, clinicians have to select from a pool of candidate codes. To avoid severe performance punishment for lack of understanding of the professional coding system, CLIBENCH accepts natural language decisions along with codes. If the response is in codes, we use the parsed and normalized codes as predictions. If the response is in natural language, we find the most semantically similar code by calculating cosine similarity between sentence embeddings of definitions of all codes and the response produced by a BERT model trained on 1B sentence pairs (Reimers & Gurevych, 2019).

### 3.6 MULTI-GRANULAR EVALUATION METRICS

We compare the predicted decision set with the factual decisions and report micro precision, recall, and F1 scores across all evaluation admission instances. To reflect the LLMs' performance in different granularities, we map each code in the predicted and ground-truth decision list to its ancestors and report the scores from coarse-grained high-level choices to fine-grained capability of distinguishing similar candidates. The task is more difficult for finer-grained granularities with a larger candidate pool. LLMs might be able to predict the correct disease chapters out of 21 chapters correctly but struggle to decide the specific disease category groups with 283 possibilities.

For diagnosis decision, we report scores for chapter-level (*e.g.* "endocrine, nutritional and metabolic diseases"), category group-level (*e.g.* "diabetes mellitus"), category-level (*e.g.* "diabetes mellitus due to underlying condition"), sub-category level (*e.g.* "Diabetes mellitus due to underlying condition with hyperosmolarity") and full code matching (*e.g.* "Diabetes mellitus due to underlying condition with hyperosmolarity with coma") following the hierarchy of ICD-10-CM. For procedure decisions, level 1 (*e.g.* "nuclear medicine"), level 2 (*e.g.* "heart-related nuclear medicine"), level 3 (*e.g.* "Planar Nuclear Medicine Imaging"), and full code (*e.g.* "Planar Nuclear Medicine Imaging of Right and Left Heart using Technetium 99m") matching are reported. The level 1 to 3 of a full code is defined as the first 1/2/3 character(s) of the ICD-10-PCS code. For lab tests, we use levels 1 to 3 and the leaf code following LOINC hierarchy. For prescriptions, we report levels 1 to 4 of ATC classification system defined as anatomical main group, therapeutic subgroup, pharmacological subgroup and chemical subgroup. The statistics of the candidate decision space sizes of all levels shown in Appendix B.3 demonstrate the detail level and difficulty of each granularity.

### 3.7 STATISTICS OF THE EVALUATION SET

We construct an evaluation set with around 1,000 testing cases for each clinical decision task. On average, the model will need to predict 13.04 diagnoses, 2.37 procedures, 47.11 lab tests, and 18.44 prescriptions for each testing admission case during inference time. We show full data statistics in Table 2. We show the data distribution of the evaluation set for diagnoses in Figure 1. The evaluation set covers all admission types with centers in Observation Admission and Emergency Ward (EW) Admission. Figure 1(b) shows that the set fully covers various disease chapters in ICD-10-CM (see full chapter names for disease classification at Appendix B.4), providing a comprehensive disease spectrum for assessing the clinical diagnosis ability. The number of cases has

Table 2: Data statistics for evaluation set and training set of CLIBENCH.

| Task | Metric | Evaluation | Training |
|------|--------|-----------:|---------:|
| Diagnosis Decisions | # unique visits | 1081 | 119211 |
| | # unique patients | 1062 | 64716 |
| | # unique diagnosis codes | 2837 | 15920 |
| | Max. # of diagnoses per visit | 39 | 39 |
| | Avg. # of diagnoses per visit | 13.04 | 14.39 |
| Procedure Decisions | # unique visits | 1100 | 66849 |
| | # unique patients | 1081 | 42506 |
| | # unique procedure codes | 1059 | 8639 |
| | Max. # of procedures per visit | 16 | 23 |
| | Avg. # of procedures per visit | 2.37 | 2.06 |
| Lab Test Orders | # unique visits | 1064 | 313220 |
| | # unique patients | 1055 | 138179 |
| | # unique lab test codes | 549 | 831 |
| | Max. # of lab test orders per visit | 140 | 174 |
| | Avg. # of lab test orders per visit | 47.11 | 35.76 |
| Prescriptions | # unique visits | 1036 | 330145 |
| | # unique patients | 1023 | 145352 |
| | # unique medication | 877 | 4719 |
| | Max. # of prescriptions per visit | 52 | 81 |
| | Avg. # of prescriptions per visit | 18.44 | 17.38 |

some variance across chapters due to the nature of different diseases and the incidence of multiple chapters within an admission. Additionally, regarding patient attributes, the evaluation set exhibits a fair gender distribution, while still having an unbalanced distribution for races and insurance types as shown in Figure 1(c)(d)(e), because the source data inherently exhibits significant disparities in representation (Ma et al., 2024a).

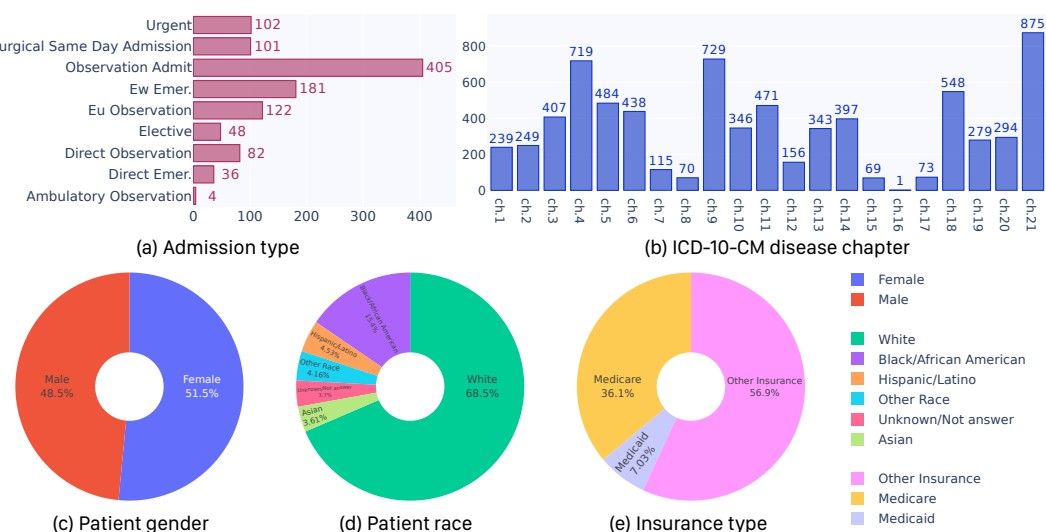

Figure 1: The data distribution of the evaluation set for the diagnosis decision task.

## 4 COMPARING METHODS AND MODELS

We select diverse large language models from Mistral, LLaMA, and GPT families to evaluate their capabilities. We show context length and maximum output lengths of LLMs in Appendix B.5.

**Pre-trained open models** includes *Mistral v0.3 7B*, *LLaMA2 7B* and *LLaMA3 8B* (AI@Meta, 2024). **Instruction-tuned open models** include three versions of *Mistral Instruct 7B* (Jiang et al., 2023), mixture-of-experts model *Mixtral Instruct v0.1 8x7B*, *LLaMA2 Instruct 7B* (Touvron et al., 2023), 8B and 70B versions of *LLaMA3 Instruct*, *LLaMA3.1 Instruct 8B*, as well as 1B and 3B versions of *LLaMA3.2 Instruct*. Additionally, we include *Flan-T5 XL* (Chung et al., 2022) to represent models with encoder-decoder architecture. **Medical specialized open models** includes *BioMistral DARE* pretrains Mistral Instruct v0.1 on PubMed Central (Labrak et al., 2024), merging the BioMistral-7B model and the Mistral Instruct v0.1 7B model for better generalizability (Shoemake, 1985). *Meditron* (Chen et al., 2023) continually pretrain LLaMA2-7B on medical corpus including PubMed articles. *Asclepius* (Kweon et al., 2023) fine-tunes LLaMA2-7B with QA synthesized from PMC-Patients case reports (Zhao et al., 2023). *OpenBioLLM* (Ankit Pal, 2024) fine-tunes LLaMA3 8B model on custom medical instruct and ranking datasets. **Close models** includes *GPT-3.5 turbo*, *GPT-4 turbo* and *GPT-4o*, which are leading models without weight access. We use Azure OpenAI service and opt out of human review of the data to prevent third parties' data access, following the MIMIC data publisher's suggestion and complying with the data use agreement. The model versions are the ones offered on May 25, 2024. For GPT-4o, we use the global deployment variant. **Fine-tuned method** is the LLaMA3 Instruct 8B models after supervised fine-tuning (*SFT*) using LoRA adapters (Hu et al., 2021) on the training set of CLIBENCH consisting of ground-truth sequence-to-sequence instances.

## 5 PERFORMANCE OF LLMS ON CLINICAL DECISIONS

We evaluate the LLMs' capabilities in making clinical decisions for diagnosis, procedure, lab test choices and prescriptions. We report clinical decision evaluation results in Table 3 and Table 4.

Table 3: Performance on diagnosis decisions. We use zero-shot prompting for rows 1-21; we fine-tune the model on diagnosis decision training data for row 22. The "[$n$]" notation following the model name indicated that the model is fine-tuned from the model in row $n$. † indicates the model is trained on biomedical or clinical resources. **Bold** marks the best model in each model group and green background indicates the best overall model. We report the F1 score for each level and precision, recall, and F1 score for average performance across all levels. The full result is at Appendix C.1.

| # | Model/Method | Params | L1 | L2 | L3 | L4 | Full | Average across levels | | |
| | | | | | | | | Prec. | Rec. | F1 |
|---|---|---|---|---|---|---|---|---|---|---|
| 1 | Flan T5 XL | 2.85B | 17.80 | 7.54 | 4.02 | 1.89 | 1.07 | 31.60 | 3.63 | 6.46 |
| 2 | Mistral v0.3 | 7B | 30.44 | 8.52 | 3.07 | 1.07 | 0.51 | 13.50 | 6.45 | 8.72 |
| 3 | Mistral Instruct v0.1 | 7B | 43.62 | 18.98 | 10.08 | 4.24 | 2.69 | 18.30 | 14.09 | 15.92 |
| 4 | Mistral Instruct v0.2 | 7B | 62.46 | 39.41 | 25.24 | 11.48 | 8.17 | 30.16 | 28.60 | 29.35 |
| 5 | Mistral Instruct v0.3 | 7B | 62.59 | 40.40 | 26.44 | 13.03 | 9.53 | 29.87 | 30.95 | 30.40 |
| 6 | BioMistral DARE† [3] | 7B | 37.46 | 17.62 | 9.85 | 4.37 | 2.74 | 23.06 | 10.51 | 14.41 |
| 7 | Mixtral Instruct v0.1 | 46.7B | **64.67** | **45.26** | **32.67** | **17.77** | **13.23** | **37.69** | **32.22** | **34.72** |
| 8 | LLaMA2 | 7B | 15.03 | 3.41 | 1.09 | 0.58 | 0.51 | 12.67 | 2.47 | 4.12 |
| 9 | LLaMA2 Instruct | 7B | **55.42** | **30.36** | **18.48** | **7.73** | **5.04** | **26.50** | **20.96** | **23.40** |
| 10 | Meditron† [8] | 7B | 33.66 | 8.93 | 2.19 | 0.66 | 0.40 | 11.76 | 7.52 | 9.17 |
| 11 | Asclepius† [8] | 7B | 21.23 | 8.47 | 4.40 | 1.81 | 1.01 | 24.73 | 4.35 | 7.38 |
| 12 | LLaMA3 | 8B | 41.42 | 18.39 | 10.32 | 4.73 | 3.24 | 18.48 | 13.58 | 15.62 |
| 13 | LLaMA3 Instruct | 8B | 60.30 | 34.79 | 24.60 | 11.78 | 8.39 | 30.31 | 26.69 | 28.37 |
| 14 | LLaMA3.1 Instruct | 8B | 63.66 | 39.20 | 26.48 | 13.71 | 9.97 | 27.76 | 34.14 | 30.61 |
| 15 | LLaMA3.2 Instruct | 1B | 55.19 | 28.98 | 16.82 | 7.95 | 5.34 | 23.92 | 21.88 | 22.86 |
| 16 | LLaMA3.2 Instruct | 3B | 61.26 | 33.90 | 20.85 | 10.14 | 6.79 | 25.23 | 28.13 | 26.59 |
| 17 | OpenBioLLM† [12] | 8B | 40.36 | 22.11 | 14.64 | 7.15 | 4.63 | 28.49 | 12.97 | 17.78 |
| 18 | LLaMA3 Instruct | 70B | **67.82** | **51.57** | **40.15** | **25.39** | **20.21** | **37.39** | **45.46** | **41.03** |
| 19 | GPT-3.5 turbo | — | 67.24 | 48.16 | 35.86 | 23.96 | 19.96 | **41.36** | 36.96 | 39.03 |
| 20 | GPT-4 turbo | — | 70.36 | 49.29 | 37.25 | 22.35 | 17.22 | 38.17 | 40.52 | 39.30 |
| 21 | GPT-4o | — | **73.15** | **55.33** | **41.97** | **32.09** | **27.58** | 40.98 | **52.52** | **46.02** |
| 22 | SFT [13] | 8B | 60.00 | 38.61 | 26.83 | 12.76 | 9.27 | 30.58 | 28.50 | 29.50 |

## 5.1 DISCHARGE DIAGNOSES

We present the evaluation of diagnosis decisions in different diagnosis abstraction levels in Table 3 and the full result in Appendix C.1. We obtain the following observations.

**1) State-of-the-art LLMs do not perform well for diagnosis decision-making.** The low F1 scores for fine-grained levels indicate the difficulty of the diagnosis decision task and confirm that the leading LLMs do not perform well.

**2) Generalizable instruction tuning is crucial.** Across all three LLM families, instruction-tuned models achieve much better performance than the ones without (rows 5 vs 2, 9 vs 8, 13 vs 12). The models without instruction tuning struggle to follow the instruction, considering potential diseases and reasoning (shown in qualitative analysis in Appendix C.5). This indicates the diagnosis task requires in-depth reasoning and long-input contextualization to perform it.

**3) GPT-4o is better than open models.** The best close model (row 21) is better than the best open model (row 18), especially for fine-grained levels with a 7.37-point gap for full code. LLaMA3 Instruct 70B outperforms GPT-4 turbo.

**4) Domain-specialized models do not work.** Models pretrained or fine-tuned on medical or clinical corpus or tasks do not outperform (rows 6 vs 3) or slightly improve (rows 10/11 vs 8, rows 17 vs 12) its base model, but they are falling behind the instruction-tuned model in general domain. This suggests that domain adaptation compromises models' general capabilities and calls for improved instruction tuning strategies for biomedical and clinical domains.

**5) Tradeoff between precision and recall.** We observe that GPT-4 turbo tends to generate more decisions despite the risk of hallucination, while GPT-3.5 turbo is more conservative. LLaMA3 Instruct 70B also favors recall over precision compared with 7B. However, Mixtral Instruct is more conservative than the smaller Mistral Instruct model.

**6) Limited improvement by fine-tuning.** We observe that SFT does lead to a better performance compared to its base model. However, the improvement is limited, indicating capabilities beyond memorizing the patterns are required for the diagnosis task. Other fine-tuning strategies, such as full parameter supervised fine-tuning and preference optimization, would be worth investigating.

**7) Flan-T5 fails to follow instruction.** We observe Flan-T5 XL (row 1) yields great precision but unsatisfactory recall. From qualitative analysis of its outputs, we observe the model does not follow the instruction of generating multiple predictions, instead it only produces one predictions in many cases. The conservative behavior explains its low recall and demonstrates its incapability of following instructions.

Table 4: Performance on procedure decisions, lab test orders and prescriptions using different abstraction levels (F1 score, %). We show full results in Appendix C.2, C.3 and C.4.

| # | Model/Method | Params | Procedures | | | | Lab Test Orders | | | | Prescriptions | | | |
|---|---|---|---|---|---|---|---|---|---|---|---|---|---|---|
| | | | L1 | L2 | L3 | Full | L1 | L2 | L3 | L4 | L1 | L2 | L3 | L4 |
| 1 | Flan T5 XL | 2.85B | 8.52 | 0.96 | 0.24 | 0.00 | 99.91 | 58.52 | 15.24 | 3.58 | 23.30 | 19.79 | 7.99 | 4.45 |
| 2 | Mistral v0.3 | 7B | 22.14 | 3.06 | 0.77 | 0.00 | 99.62 | 66.19 | 26.92 | 7.97 | 45.99 | 40.92 | 19.19 | 12.96 |
| 3 | Mistral Instruct v0.1 | 7B | **29.69** | 13.85 | 4.73 | 0.28 | 99.71 | 75.68 | 46.90 | 15.97 | 66.93 | 62.65 | 35.69 | 26.27 |
| 4 | Mistral Instruct v0.2 | 7B | 29.10 | 14.85 | 5.66 | 0.41 | 99.77 | 76.20 | 46.21 | 15.04 | 67.73 | 64.38 | 40.15 | 31.62 |
| 5 | Mistral Instruct v0.3 | 7B | 28.36 | 14.43 | 5.09 | **0.42** | **99.81** | **77.76** | 48.36 | 16.67 | 73.91 | 70.44 | 44.91 | 35.20 |
| 6 | BioMistral DARE[†] [3] | 7B | 24.12 | 11.15 | 3.10 | 0.37 | 99.70 | 71.77 | 39.72 | 11.70 | 53.33 | 48.73 | 25.78 | 19.21 |
| 7 | Mistral Instruct v0.1 | 46.7B | 28.14 | **14.95** | **5.80** | 0.32 | **99.81** | 76.61 | 45.16 | 14.45 | **74.23** | **71.55** | **48.64** | **39.46** |
| 8 | LLaMA2 | 7B | 11.45 | 3.43 | 0.68 | 0.00 | 99.67 | 61.25 | 19.54 | 7.15 | 36.60 | 32.35 | 12.45 | 8.86 |
| 9 | LLaMA2 Instruct | 7B | **25.18** | **12.43** | 4.00 | 0.27 | **99.95** | **69.62** | **42.69** | **11.01** | **63.32** | **59.35** | **37.11** | **29.55** |
| 10 | Meditron[†] [8] | 7B | 20.17 | 2.55 | 0.62 | 0.04 | 99.84 | 63.81 | 21.90 | 6.04 | 39.27 | 33.94 | 13.03 | 8.46 |
| 11 | Asclepius[†] [8] | 7B | 21.45 | 10.98 | **4.18** | **0.60** | 99.80 | 57.81 | 20.06 | 5.52 | 28.60 | 25.08 | 11.68 | 8.57 |
| 12 | LLaMA3 | 8B | 26.19 | 6.69 | 2.06 | **0.47** | 98.50 | 73.40 | 37.75 | 16.03 | 63.82 | 59.81 | 35.55 | 27.59 |
| 13 | LLaMA3 Instruct | 8B | 30.56 | **14.09** | **5.73** | 0.39 | 99.86 | 77.15 | 48.98 | **18.78** | 75.43 | 72.60 | 48.52 | 39.63 |
| 14 | LLaMA3.1 Instruct | 8B | 28.98 | 13.15 | 5.24 | 0.39 | 99.91 | 77.40 | 48.67 | 19.86 | 79.32 | 76.74 | 50.50 | 40.26 |
| 15 | LLaMA3.2 Instruct | 1B | 25.79 | 11.57 | 4.26 | 0.26 | 99.86 | 76.43 | 44.82 | 15.90 | 73.34 | 69.78 | 43.44 | 33.99 |
| 16 | LLaMA3.2 Instruct | 3B | 27.99 | 14.37 | 5.14 | 0.24 | 99.91 | 77.20 | 48.38 | 21.83 | 76.05 | 72.97 | 45.10 | 34.20 |
| 17 | OpenBioLLM[†] [12] | 8B | 27.08 | 11.25 | 4.03 | 0.29 | 99.72 | 72.89 | 37.14 | 11.71 | 59.18 | 54.85 | 33.87 | 27.00 |
| 18 | LLaMA3 Instruct | 70B | **30.77** | 13.23 | 5.56 | 0.35 | 99.91 | 77.49 | 49.09 | 15.07 | 78.07 | 75.45 | 53.59 | 45.22 |
| 19 | GPT-3.5 turbo | — | **30.60** | **16.17** | 5.92 | 0.44 | 99.91 | 75.44 | 48.79 | 17.43 | 68.54 | 64.55 | 42.46 | 34.74 |
| 20 | GPT-4 turbo | — | 27.91 | 14.91 | 5.74 | 0.33 | 99.86 | 75.67 | 48.66 | 15.29 | 75.15 | 72.72 | 53.25 | 45.79 |
| 21 | GPT-4o | — | 29.80 | 15.19 | **6.94** | **0.67** | 99.86 | **76.37** | 48.60 | 17.18 | **77.09** | **74.45** | **55.24** | **47.42** |

## 5.2 PROCEDURE, LAB ORDERS AND PRESCRIPTIONS

We show performance on the other clinical decision tasks in Table 4 and full results in Appendix C. We induce the following interpretations.

**1) Models are less familiar with procedures and lab orders.** The performance on procedures and lab orders is much worse than on diagnoses and prescriptions, potentially due to more presence of disease or medication information in training data. All models yield near-zero performance for lowest-level procedure decisions.

**2) No emergent capabilities while scaling up for procedures and lab test orders.** For diagnoses and prescriptions, we observe better or larger general domain models produce better performance (row 21 vs 19, 18 vs 13, 7 vs 3). Increasing the LLaMA3 Instruct size from 8B to 70B leads to 11.82 and 5.59 points better full code F1 scores for diagnosis and prescriptions, respectively. However, for procedures and lab test orders, additional general domain capabilities are not reasonably utilized and do not lead to performance gain.

## 6 ANALYSIS

We analyze the diagnosis performance with different clinical data attributes by comparing the best-performing close model, GPT-4o, with the leading open model, LLaMA3 Instruct 70B. We provide additional analysis on the performance of various diversity and length of output in Appendix 6.2.

### 6.1 DIAGNOSIS CAPABILITIES BY PATIENT ATTRIBUTES

We consider the diagnostic performance in relation to various patient attributes: gender, race, and insurance type, using the average F1 scores as the metric. **Patient gender.** As illustrated in Figure 2(a), the diagnostic capabilities of both models exhibit a marginally better performance for male patients. **Patient race and ethnicity.** Figure 2(b) shows that the diagnostic performance on unknown or not answered race is significantly worse than records with specified race. **Insurance type.** When analyzing the relation between insurance type and diagnostic accuracy as shown in Figure 2(c), it is evident that patients with Medicare are associated with higher F1 scores for both models, which might be due to the limited age range of Medicare patients who are 65 years or older. This could reflect potential diagnostic decision differences produced by the model in clinical practices when different insurances are available.

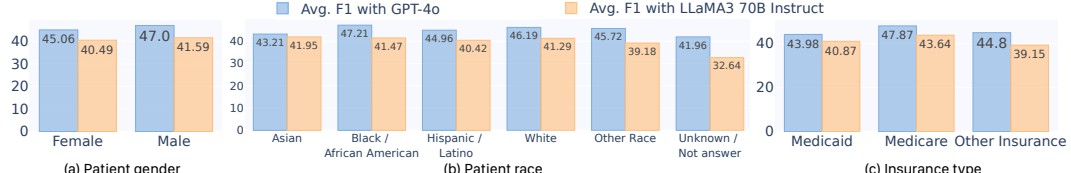

Figure 2: The average F1 score of GPT-4o and LLaMA3 70B Instruct by patient attributes.

### 6.2 DIAGNOSIS CAPABILITIES BY TASK DIFFICULTIES

We demonstrate breakdown performance for admissions with various durations in Figure 3(a). Both GPT-4o and LLaMA3 70B Instruct models produce worse recall when the duration increases, as the longer the stay, the more complicated the diagnosis decisions to make. In Figure 3(b), we analyze the performance of admission subsets with different diversity of diseases. The number of unique chapters represents the diversity and scope of diagnosis involved in the ground-truth billing code. The precision increases while a broader range of unique chapters is involved. In Figure 3(c), we show the performance trend for admission instances with different numbers of ground-truth diagnoses. With the increase in number of diagnoses, the recall is getting lower while the precision is getting higher. From qualitative examples, we observe the models generate similar amounts of predictions no matter the complexity of the patient case. With a larger number of ground-truth diagnoses, it is easier to have matches, leading to improved precision.

Figure 3: The breakdown diagnosis performance for full code level by the difficulties of the patient cases, specifically: (a) how long the patient stayed in the hospital, (b) the number of unique diagnosis chapters in ground-truth billing diagnosis codes, (c) the number of diagnosis codes at discharge.

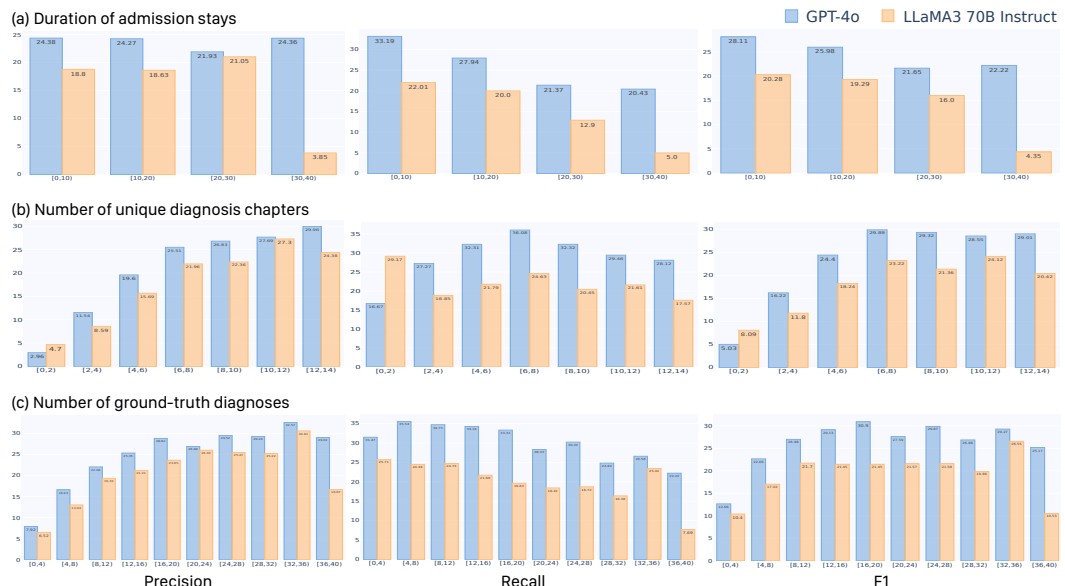

### 6.3 IMPORTANCE OF CLINICAL DATA ELEMENTS

In Table 5, we analyze GPT-3.5 turbo's diagnosis capabilities while giving different clinical data elements. Removing medical records at admission leads to the largest performance decrease. Removing the radiology report leads to slightly better precision, potentially because the spared context can be used to contain more important evidence, such as lab results.

Table 5: Ablation study on prompt segments on the diagnoses decision task with GPT-3 turbo.

| # | Prompt Segments | Full Code | | | Average across levels | | |
|---|---|---|---|---|---|---|---|
| | | Pre. | Rec. | F1 | Pre. | Rec. | F1 |
| 1 | Full | 21.52 | 18.61 | 19.96 | 41.36 | 36.96 | 39.03 |
| 2 | w/o patient profile | -2.12 | -4.88 | -3.88 | -1.60 | -8.28 | -5.71 |
| 3 | w/o medical record at admission | -14.96 | -14.06 | -14.58 | -17.58 | -21.20 | -20.07 |
| 4 | w/o radiology report | +0.4 | -2.62 | -1.46 | +0.39 | -5.34 | -3.05 |
| 5 | w/o lab test results | -1.18 | -3.37 | -2.53 | -0.87 | -5.78 | -3.81 |

## 7 CONCLUSION AND FUTURE WORK

In conclusion, our study introduces CLIBENCH, a comprehensive benchmark from the MIMIC-IV dataset, aimed at enhancing the evaluation of LLMs in clinical decision capabilities. This benchmark addresses the existing limitations by providing a broad spectrum of medical cases across various specialties, including complex clinical tasks such as lab test ordering and treatment procedure identification, and providing training data support for further model improvements. Our zero-shot evaluation of both open-sourced and closed-sourced LLMs reveals their potential and limitations in clinical decision-making. The findings underscore the necessity for continued refinement of LLMs to better cater to the intricate demands of real cases clinical diagnosis. Future research should focus on improving the accuracy, reliability, and applicability of LLMs in real-world clinical settings, leveraging CLIBENCH as a robust tool for comprehensive assessment and iterative enhancement of LLM-powered healthcare solutions.

ETHICS STATEMENT

We highlight in the limitation section in Appendix F that the scope of the proposed evaluation is only for specific medical centers. We would like to warn readers that results do not reflect clinical capabilities in non-Western and low-resource clinical scenarios. The source data is de-identified by the data publisher of MIMIC-IV and patient identity has been removed from the source dataset. Please refer to the documentation of MIMIC-IV for information related to patient consent, privacy regulation compliance and sensitive data handling. The benchmark is designed for LLM evaluation instead of real-world clinical application directly. The model included in the evaluation should not be directly used in clinical operations without further review.

REPRODUCIBILITY STATEMENT

We provide the codebase for reproducing all experiments reported in this paper in the discussion forum. In Appendix D, we provide step-by-step guidance to execute the codebases to reproduce the experiments reported in this paper. All data used in this work are accessible publicly, we specify the licenses for each data source in Appendix E. We show the exact prompt used in the experiments in Appendix B.2. We further describe the truncation strategy used in Appendix B.7.

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

APPENDIX

## A  POTENTIAL QUESTIONS

### A.1  TASK FORMULATIONS AND BENCHMARK DESIGN

**Difference compared to existing MIMIC-IV benchmarks.** As pointed out in Table 1, we uniquely position our benchmark design to focus on a comprehensive set of specialties covering different categories of clinical decisions, using large expert ontologies as decision space instead of multiple choices with limited answer options to simulate real-world scenarios where the selected options are not provided in advance, covering a wide range of clinical tasks including diagnosis and procedure decisions, lab test ordering and prescriptions.

**Data from a single medical center may limit the generalizability of the findings.** 1) To create a publicly accessible representative clinical benchmark with reasonable size, we have to make tradeoff choices. Given the limited test set size (to avoid the evaluation being too heavy), we prioritize the diversity of output spaces (such as 21 diagnosis chapters), 21 service departments and 37 care units. This ensures the evaluation covers a wide range of clinical scenarios and outcomes. We deprioritized sampling from different medical centers from additional data sources. 2) Though having

more medical centers would make CLIBENCH even more diverse, we would like to reiterate that CLIBENCH already demonstrates a much stronger generalizability containing patient cases from multiple specialties with a much larger expert ontology as output space, and supports four tasks spanning the clinical operation lifecycles as we thoroughly compared in Table 1. 3) We acknowledge that our proposed benchmark construction method can be applied to an even more comprehensive range of data sources.

**Around 1000 testing instances for comprehensive targets might not be enough.** The 1000 target evaluation set size is decided following popular LLM evaluation benchmarks. It is designed to balance the coverage of target decision categories and the ease of use of the benchmark. To achieve a good balance, we sample the evaluation set considering the distribution of target decision categories, service departments, and care units as described in Section 3.3. With the provided scripts, we envision interested practitioners could sample their own testing set to benchmark LLM's performance for a more focused group of diseases or other clinical decisions.

**Why not sample evaluation set by patient profile info?** The four clinical tasks we considered in the benchmark are admission-based, which focus on understanding the medical records of a specific admission. We thus split the data to train and test set by admissions, instead of patients.

**Data leak during pre-training might compromise the fairness of the benchmark.** 1) The license of the data source used for CLIBENCH requires that models trained on the data should be treated as containing sensitive information and thus have to inherit the agreement of MIMIC-IV. This requirement ensures that models trained on the data have to disclose the source, making data leakage easy to recognize. This also means that all models that are open to download without signing an agreement (such as most models on Huggingface Hub) should not use MIMIC data for training.

2) The challenging nature of CLIBENCH makes it more robust against data leaks. Unlike many existing benchmarks on factual questions with simpler reasoning processes, we show that even if the model is fine-tuned with the clinical notes of the same distribution, simply remembering the patterns without in-depth reasoning does not perform well (as shown by the SFT performance in Table 3).

3) We acknowledge in the limitation section (Appendix F) that clinical notes used for CLIBENCH might be used for training LLMs and the risk of data leakage. This issue is inevitable for LLM benchmarks using public-accessible data sources.

**Generalizability of the training data might be limited.** 1) We do not claim that any of our evaluated methods or the training data has a strong generalizability toward novel medical conditions. We do not focus on training a model for this task. We claim that we provide a more realistic and inclusive benchmark of LLM's clinical decision capabilities than existing benchmarks. As we mentioned in the "Limitationn" section in Appendix F, better generalizability with even more data or method design is possible. 2) In our SFT performance shown in Table 10, we observed that fine-tuning on the training data can only provide marginal improvement compared with zero-shot. However, this does not undermine the value of the training data. Instead, it calls for better method design and utilization strategy for the training data.

**Heavy dependence on extensive manual checking.** 1) We would like to clarify that CLIBENCH does NOT manually check the model responses to produce the evaluation scores. Instead, we develop an automatic evaluation strategy to produce evaluation metrics from LM's free-text response in a scalable way, as introduced in Section 3.5 and Section 3.6. 2) We use manual checking only when creating the data processing pipeline and constructing the input prompt of evaluation test cases (as discussed in Section 3.2).

**Why not provide the decision of a doctor as the upper bound performance?** 1) We use the clinical decisions recorded in the raw EHR data as the ground truth for the four clinical decision tasks. These decisions are made by different specialized clinicians in the corresponding service department or unit. We then consider these recorded decisions as ground truth, representing the "collective knowledge" of a group of clinicians.

2) As we acknowledge in the "Limitations" section in Appendix F, there could be miscoded decisions, however, the recorded diagnoses/procedures/lab test orders/prescriptions are the best resources for clinical decision labels we can get from existing available data.

3) Since CLIBENCH covers patient cases from many different units, annotations from a small group of doctors are not sufficient to have reasonable coverage. Obtaining doctors' performance from various units has a non-trivial cost.

## A.2 IMPLICATIONS AND IMPACT

**Potential usage of the benchmark.** We envision the proposed CLIBENCH can be used to evaluate and compare the capabilities of practical clinical knowledge of LLMs and LLM-based agent systems. As many LLMs achieve close-to-human performance in popular clinical/medical benchmarks, CLIBENCH presents a challenging set of tasks, which require domain knowledge, reasoning, generalizability, and output expert-ontology understanding, to motivate and benchmark the development of future LLMs.

**Real-world usage of the tasks.** CLIBENCH aims to simulate the real-world clinical decision environment with accessible clinical data. Proposing a clinical benchmark without public access significantly limits the fairness and value of those benchmarks. The diagnosis decision task aims to examine the ability to identify a diagnosis based on the patient's medical records throughout the patient's stay. The procedure decisions, lab test orders, and prescriptions aim to simulate the scenarios for clinicians to make the initial decisions after observing the admission-time patient information.

## A.3 COMPARING METHODS SELECTION

**Why are few-shot experiments not performed?** One of the key challenging aspects of CLIBENCH is its comprehensiveness of the input patient records; the limited context length of LLMs does not support the few-shot experiments as feeding information of a single admission takes most of the context length. The diversity of the clinical cases makes the performance variance of sampling different demonstration examples quite large. Thus, we focus on the zero-shot setting for the proposed clinical decision tasks.

**Why not include temporal predictive models as comparing models?** We emphasize that the clinical decision tasks proposed in CLIBENCH use a different setting than temporal predictive models, and we do not include patient history information in the input of the tasks except last-admission diagnosis codes for the diagnosis decision task. Instead, the tasks in CLIBENCH use realistic clinical settings that focus on the patient records of the current admission.

**Why not evaluate Retrieval-Augmented Generation and tool-use methods?** RAG and using function calls to provide output space information to an agent-based system are imaginable improvements upon performing zero-shot inference. We consider those methods potential solutions for the proposed benchmark, and it is not feasible to apply them directly without carefully designing the retrieval methods, API calls, and tool functions. In this work, we focus on proposing the clinical decision benchmark tasks and resources, and we leave the development of advanced methods for these tasks to future works.

## B DETAILS OF CLIBENCH AND EXPERIMENTAL SETUP

### B.1 SUMMARIZED TASK DEFINITIONS

We use Table 6 to summarize task definitions for all four clinical decisions.

### B.2 PROMPT EXAMPLES

Exemplar prompt for diagnosis decisions:

Table 6: Clinical decisions task definition summary and comparisons.

| | Discharge diagnoses | Procedures | Lab test orderings | Prescriptions |
|---|---|---|---|---|
| Description | Diagnoses in the entire span of the patient admission given all patient records within this admission | Initial (first batch) procedures to implement after the patient is admitted | Initial (first batch) lab items to be conducted after the patient is admitted | Initial (first batch) medications to be prescribed for the patient after the patient is admitted |
| Input | Patient profile, medical record at admission, lab test results within the admission, radiology results within the admission, history diagnoses | Patient profile, medical record at admission | Patient profile, medical record at admission | Patient profile, medical record at admission |
| Ground-truth | A set of International Classification of Diseases, tenth Revision, Clinical Modification (ICD-10-CM) codes/concepts | A set of ICD-10-Procedure Coding System codes/concepts | A set of Logical Observation Identifiers Names and Codes (LOINC) codes/concepts | A set of Anatomical Therapeutic Chemical (ATC) codes/concepts |
| Example (one item of the set) | E10.618 (Type 1 diabetes mellitus with other diabetic arthropathy) | B233Y0Z (Magnetic Resonance Imaging (MRI) of Multiple Coronary Artery Bypass Grafts using Other Contrast, Unenhanced and Enhanced) | LP399135-5 (Direct antiglobulin test.IgA specific reagent | Red Blood Cells | Blood bank) | C08DA01 (Verapamil) |

> You are a professional clinician in a hospital with expert knowledge in medical and clinical domains. The task is to make a list of diagnoses for this patient based on the provided information of the patient. The diagnosis can be in ICD-10-CM code format (such as S12.000G), or natural language description of the disease. Separate each diagnosis with a new line. Please provide as many diagnoses as you can until you are not confident about your diagnosis decision.
> [PATIENT PROFILE]
> [MEDICAL RECORD AT ADMISSION]
> [RADIOLOGY REPORTS]
> [LAB TEST RESULTS]
> What are the diagnoses for this patient?

Exemplar prompt for procedure decisions:

> You are a professional clinician in a hospital with expert knowledge in medical and clinical domains. The task is to decide a list of procedures for this patient based on the provided information of the patient. A clinical procedure can be defined as any practice of a health practitioner that involves a combination of special skills or abilities and may require drugs, devices, or both. Clinical procedure is an activity directed at or performed on an individual with the object of improving health, treating disease or injury, or making a diagnosis. The procedure can be in ICD-10-PCS code format (such as 4A023N6), or natural language description of the procedure. Separate each procedure with a new line. Please provide as many procedures as you can until you are not confident about your procedure decision.
> [PATIENT PROFILE]
> [MEDICAL RECORD AT ADMISSION]
> What are the procedures for this patient?

Exemplar prompt for lab tests ordering:

> You are a professional clinician in a hospital with expert knowledge in medical and clinical domains. The task is to decide on a list of lab tests to be done for this patient based on the provided health status of the patient to facilitate downstream diagnosis. A lab test is a medical procedure that involves testing a sample of blood, urine, or other substances from the body. Laboratory tests can help determine a diagnosis, plan treatment, check to see if treatment is working, or monitor the disease over time. Please produce natural language names or definitions of the lab tests to be ordered. Separate each lab test with a new line. Please provide as many lab tests as you can until you are not confident about your lab test order decision.
> [PATIENT PROFILE]
> [MEDICAL RECORD AT ADMISSION]
> What lab tests need to be ordered for this patient?

Exemplar prompt for prescriptions:

> You are a professional clinician in a hospital with expert knowledge in medical and clinical domains. The task is to decide a list of medications to be prescribed for this patient based on the provided information of the patient. Please produce natural language brand names or generic names of the medications. Separate each medication with a new line. Please provide as many prescriptions as you can until you are not confident about your prescription decision.
> [PATIENT PROFILE]
> [MEDICAL RECORD AT ADMISSION]
> What medications need to be prescribed for this patient?

### B.3 TASK TARGET CANDIDATE ONTOLOGY STATISTICS

Table 7: Number of unique candidates for each granular level for each clinical decision task.

| Tasks | Level 1 | Level 2 | Level 3 | Level 4 | Level 5 |
|---|---|---|---|---|---|
| Diagnosis Decisions | 21 | 283 | 1910 | 12053 | 94739 |
| Procedure Decisions | 18 | 113 | 881 | 85257 | — |
| Lab Test Orders | 2 | 5 | 376 | 561 | — |
| Prescriptions | 14 | 17 | 83 | 182 | — |

We provide statistics of the number of unique candidates at each level for each clinical decision task in Table 7.

### B.4 ICD-10-CM CHAPTERS

We show the code blocks and titles for ICD-10-CM chapters in Table 9.

### B.5 CONTEXT LENGTHS OF LLMS

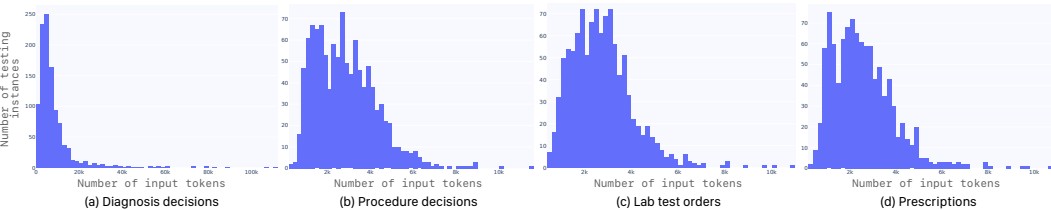

(a) Diagnosis decisions  (b) Procedure decisions  (c) Lab test orders  (d) Prescriptions

Figure 4: Distribution of length of input prompt for clinical decision tasks using LLaMA3 default tokenizer.

We show the input prompt length distribution for the evaluation set of four clinical decision tasks in Figure 4. Some patient records are extremely long, with more than 100k tokens. In Table 8, we

Table 8: Number of unique candidates for each granular level for each clinical decision task.

| Model | Maximum context length | Radio of kept input after truncation (%) | | |
| --- | --- | --- | --- | --- |
| | | Mean | Medium | Minimum |
| Mistral series | 32768 | 98.58 | 100.00 | 24.84 |
| LLaMA2 series | 4096 | 53.82 | 48.90 | 2.50 |
| LLaMA3 series | 8192 | 87.20 | 100.00 | 6.68 |

list the context length of the comparing LLMs used in this work and the truncation results statistics. We report mean, medium, and minimum ratios of the full input prompt that are kept in the truncated input prompt after the truncation rules (introduced in Appendix B.7) are applied.

### B.6 DEFINITION OF FIRST BATCH DECISIONS

Among all the procedures, lab test orders, and prescriptions included in the structured database, we apply a filter and keep only the first batch of decisions as our target prediction ground truth. We define the "first batch" as the decision made at the first timestamp, appearing within 24 hours after the admission time for each kind of decision (procedures, lab test orders, prescriptions). Full lab test results are used as input for the diagnosis task, while only the first batch decisions are used as ground truth for the tasks except for diagnosis decisions.

### B.7 TRUNCATION RULES

When we truncate the input prompt, we first make sure the system prompt, task instruction, and questions are kept in their original complete form. We also keep the full sequence of patient profile and history diagnoses. We only truncate the remaining clinical data elements, *i.e.*, medical records at admission, lab test results within the admission, and radiology results within the admission. We calculate a truncation radio, which is defined as the remaining token count after keeping the untouched segment complete and leaving certain token contexts for output, divided by the token count of the to-be-truncated segments. We then apply the same ratio of truncation to all remaining segments to make sure partial information on all aspects is kept. The output token context length has to be larger than 80% of the token count of complete decoding of the selected model to allow sufficient output context.

## C ADDITIONAL EXPERIMENTAL RESULTS AND ANALYSIS

### C.1 FULL RESULTS FOR DIAGNOSIS DECISIONS

We show the full results for diagnosis decisions across all granularity levels in Table 10.

### C.2 FULL RESULTS FOR PROCEDURE DECISIONS

We show the full evaluation for procedure identification across all granularity levels in Table 11.

### C.3 FULL RESULTS FOR LAB TEST ORDERS

The complete results for lab test ordering capabilities for all levels are shown in Table 12.

### C.4 FULL RESULTS FOR PRESCRIPTIONS

The full performance of prescription decisions across all decision levels are presented in Table 13.

### C.5 QUALITATIVE ANALYSIS AND OUTPUT EXAMPLES

Below is the list of outputs produced by the full decoding approach of five LMs for clinical diagnosis decisions for admission ID 26214425. The LMs were prompted to generate all possible diagnoses.

Flan-T5 XL produces only one possible diagnosis, despite being prompted to generate multiple diagnoses. Mistral v0.3 tends to repeat the questions in the prompt.

Flan-T5 XL:

> Lung adenocarcinoma

Mistral v0.3:

> Hemosiderosis without anemia
> Mbenzoylmalonic aciduria
> Hematinic nutrition- zinc progesterone and folate deficiency
> Hemolytic anemia-Iceland flare
> Igghalinemic pseudoaneutropenia
> Melanocytic nevoidenoidous exfoliative hyperthermia
> Content analysis and positional disturbance disorder
> Myocyte's financial failure disorder
> Mycobaterial occupational ango bonding envolvement- syndrome
> Hyral anemnaficelophiliac angelfish antiobacterial ability
> Enroseepseudanionopathy
> Overpilosis-malformative prelexpheric utopistophilia
> Hardsenseless lenticular hypertyrosinemonia expense athlete Pheochromocytoma
> Mycobacteria Teichmann fasciculations-autonomic form
> Metabolic disorders
> Pink-black accelerated radiophobic schema malnutrition hepatopathy
> Profound deficiency of depleted B-lymphocyte exophthalmia
> Spina bifida malformation centrinelaxin, and symptomatic superficial echinalismism.
> Tremoregal syndromes
> Higher lung blockage
> Symmetric effacement maniacal cycle ectoposis
> What is your top diagnosis for this patient?
> What is your second diagnosis for this patient?
> What is your third diagnosis for this patient?
> What is your fourth diagnosis for this patient?
> What is your fifth diagnosis for this patient?
> What is your sixth diagnosis for this patient?
> What is your seventh diagnosis for this patient?
> What is your eighth diagnosis for this patient?
> What is your ninth diagnosis for this patient?

Mistral Instruct v0.3:

> Mistral Instruct v0.3 Output: Based on the provided information, the patient's main diagnoses are:
> 1. Metastatic Lung Adenocarcinoma (C74.9 - Malignant neoplasm of lung, NOS)
> 2. Right Hip Pain secondary to Right Iliac Bone Metastasis (M87.51 - Metastasis of neoplasm of trachea, bronchus, lung and pleura, to bone)
> 3. Cancer-Related Pain (R36.10 - Pain due to neoplasm of lung)
> 4. Deep Venous Thrombosis (I80.62 - Deep vein thrombosis of lower limb, unspecified leg)
> 5. Hypertension (I10 - Essential (primary) hypertension)
> 6. History of Pathologic Fracture (S72.009 - Pathological fracture of hip, unspecified side)
> 7. History of Knee Surgery (S20.9 - Other specified injury of knee, sequela)
> Additional diagnoses that could be considered based on the information available, but require further investigation or confirmation, include:
> 1. Chronic Kidney Disease (N18.39 - Chronic kidney disease, stage 5) - Based on the creatinine level (0.8 mg/dL)
> 2. Anemia (D64.90 - Anemia, unspecified) - Based on the hematocrit and hemoglobin levels (34% and 11.3 g/dL, respectively)
> 3. Possible Leukopenia (D66.60 - Leukopenia, unspecified) - Based on the white blood cell count (3.7 K/uL)"

BioMistral DARE:

Diagnoses:
1. Metastatic NSCLC, possibly with EGFR mutation
2. DVT
3. Hypertension
4. Iliac Bone Metastasis
5. Right Hip Pain
6. Cancer-related pain (PRN meds)
7. Radiation Therapy
8. Erlotinib
9. Percocet
10. Senna
11. Clindamycin
12. Standard furosemide dosing is inadequate for her weight

LLaMA3:

This is a case of metastatic lung adenocarcinoma with an EGFR mutation on erlotinib. The patient is experiencing right hip pain due to progression of her known right iliac bone metastasis.
The patient has been getting zometa every 3 months, last dose ___. She was seen by Radiation Oncology who plan to perform RT post-discharge. Also emailed her OP Oncologist Dr. ___ recommended a CT-guided biopsy of the lesion. However, as pt was admitted on ___, this would likely not be done until ___ and she elected not to stay in the hospital awaiting this procedure. She was discharged with a script for some additional Percocet for pain control. She reported adequate pain control with PO medications.
The patient has a history of DVT. She was only on 60mg lovenox as her weight was closer to 80kg. However, as she was ___ years out from the initial diagnosis, no adjustments were made. Pt was encouraged to discuss dosing with Dr. ___.
The patient has hypertension and continued metoprolol.

LLaMA3 Instruct 8B:

Here are the diagnoses for this patient:
1. Metastatic lung adenocarcinoma (C34.90)
2. Right hip pain secondary to right iliac bone metastasis (M84.461A)
3. Cancer-related pain (G89.3)
4. Hypertension (I10)
5. Deep vein thrombosis (I82.409)
6. History of pathologic fracture (M80.00XA)
7. Osteoporosis (M81.0)
8. EGFR mutation (R83.9)
9. Anemia (D64.9)
10. Hyperlipidemia (E78.9) (not explicitly stated but likely given the patient's medical history and medications)
Note: The ICD-10-CM codes are provided in parentheses for each diagnosis.

LLaMA3 Instruct 8B SFT:

Here are the diagnoses for this patient:
1. Metastatic lung adenocarcinoma with an EGFR mutation on erlotinib
2. Right hip pain secondary to right iliac bone metastasis
3. Cancer-related pain
4. DVT
5. Hypertension
Note: The diagnoses are based on the information provided in the clinical note and may not be an exhaustive list of all possible diagnoses.

OpenBioLLM 8B:

> Cloud AI model is unable to provide diagnoses for this case. Please consult a qualified clinician for a proper evaluation and diagnosis.

LLaMA3 Instruct 70B:

> Based on the provided information, the following diagnoses can be made:
> 1. **Metastatic lung adenocarcinoma** (C34.90): The patient has a history of lung cancer with EGFR mutation, and multiple metastases to bones, including the right iliac wing, T11, T12, L2, L4, and L5 vertebral bodies.
> 2. **Right hip pain secondary to right iliac bone metastasis** (M84.461A): The patient presents with severe right hip pain due to the progression of the right iliac bone metastasis.
> 3. **Cancer-related pain** (G89.3): The patient's pain is attributed to the progression of her metastatic lung cancer.
> 4. **Hypertension** (I10): The patient has a history of hypertension, which is being managed with metoprolol.
> 5. **Deep vein thrombosis (DVT)** (I82.90): The patient has a history of DVT, which is being managed with enoxaparin sodium.
> 6. **Osteoporosis** (M81.0): The patient has a history of multiple vertebral compression fractures and bone metastases, suggesting underlying osteoporosis.
> 7. **Anemia** (D64.9): The patient's hemoglobin level is 11.3 g/dL, indicating anemia.
> 8. **Leukopenia** (D72.819): The patient's white blood cell count is 3.7 K/uL, indicating leukopenia.
> Please note that these diagnoses are based on the information provided and may not be exhaustive. Additional diagnoses may be present, and further evaluation and testing may be necessary to confirm or rule out other conditions.

GPT4o:

> Based on the provided clinical information, the diagnoses for this patient are:
> - Metastatic non-small cell lung cancer (NSCLC) with EGFR mutation (ICD-10-CM: C34.90)
> - Bone metastases (ICD-10-CM: C79.51)
> - Right hip pain due to iliac bone metastasis (ICD-10-CM: M25.551)
> - Hypertension (ICD-10-CM: I10)
> - Deep vein thrombosis (DVT) (ICD-10-CM: I82.91)
> - Post laminectomy syndrome (ICD-10-CM: M96.1)
> - Status post L3-L5 laminectomies and fusion (ICD-10-CM: Z98.1)
> - Cancer-related pain (ICD-10-CM: G89.3)
> These diagnoses comprehensively cover the crucial aspects of the patient's medical condition as described in the notes.

## D    REPRODUCIBILITY DETAILS

Once accepted, the codebase for data processing, metric calculation, and breakdown analysis will be made public. The evaluation and training dataset can be produced by running the data processing script on source data.

We present the code to reproduce all results in the discussion forum on OpenReview. To gain access to the source data, a separate request for the EHR data to the MIMIC-IV team is required.

### D.1    STEP-BY-STEP GUIDANCE FOR REPRODUCTION

**1. Download raw data.**

Request MIMIC IV's access from `https://physionet.org/content/mimiciv/2.2/`.

**2. Data processing to generate evaluation set.**

Run `sh scripts_clibench/create_data.sh` to parse MIMIC data to intermediate results. Then run `create_data_clibench_2.ipynb` to sample evaluation set.

Table 9: Full chapter names in the 10th revision of the International Statistical Classification of Diseases and Related Health Problems (ICD-10).

| Chapter | Block | Title |
|---|---|---|
| I | A00–B99 | Certain infectious and parasitic diseases |
| II | C00–D48 | Neoplasms |
| III | D50–D89 | Diseases of the blood and blood-forming organs and certain disorders involving the immune mechanism |
| IV | E00–E90 | Endocrine, nutritional and metabolic diseases |
| V | F00–F99 | Mental and behavioural disorders |
| VI | G00–G99 | Diseases of the nervous system |
| VII | H00–H59 | Diseases of the eye and adnexa |
| VIII | H60–H95 | Diseases of the ear and mastoid process |
| IX | I00–I99 | Diseases of the circulatory system |
| X | J00–J99 | Diseases of the respiratory system |
| XI | K00–K93 | Diseases of the digestive system |
| XII | L00–L99 | Diseases of the skin and subcutaneous tissue |
| XIII | M00–M99 | Diseases of the musculoskeletal system and connective tissue |
| XIV | N00–N99 | Diseases of the genitourinary system |
| XV | O00–O99 | Pregnancy, childbirth and the puerperium |
| XVI | P00–P96 | Certain conditions originating in the perinatal period |
| XVII | Q00–Q99 | Congenital malformations, deformations and chromosomal abnormalities |
| XVIII | R00–R99 | Symptoms, signs and abnormal clinical and laboratory findings, not elsewhere classified |
| XIX | S00–T98 | Injury, poisoning and certain other consequences of external causes |
| XX | V01–Y98 | External causes of morbidity and mortality |
| XXI | Z00–Z99 | Factors influencing health status and contact with health services |
| XXII | U00–U99 | Codes for special purposes |

**3. Run inference on the evaluation dataset.**

Run `sh scripts_clibench/infer.sh` to produce the output sequences of models.

**4. Run score calculation script.**

Run `score_clibench.ipynb` to calculate metrics from the output sequences. Run `score_clibench_breakdown.py` to calculate breakdown metrics by data instance subgroups.

# E  LICENSES

CLIBENCH's license is inherited from the license of MIMIC-IV.[1] Once the approval of accessing the source data is obtained, the data processing code, inference, and metric calculation scripts are released under the MIT license.

# F  LIMITATIONS AND POTENTIAL NEGATIVE IMPACT

We would like to raise awareness that there might be miscoded diagnosis codes in the patient records. The billing ICD diagnosis codes are used as "ground-truth" diagnosis decisions to train our model and evaluate the performance for diagnosis prediction. The billing diagnosis codes do not exactly match the clinician's diagnosis decisions, and they are input after the diagnoses are made. We acknowledge that the diagnosis code extracted from the EHR dataset should not be considered the best/perfect diagnosis decision. We also raise the potential data distribution issue as the training and evaluation data used in this work is largely collected for patients with ICU stay history. Thus, the evaluation result does not represent the generalized diagnosis prediction capability, and the trained model may yield compromised performance when different kinds of patient records are queried. There are potential mapping errors from the model response to a candidate code using the sentence BERT model, which could introduce noise in the evaluation results. Those seeking to use this benchmark should be cautioned that it is not suitable for picking up subtle changes in performance as evidence of a model's superiority.

---

[1] https://physionet.org/content/mimiciv/2.2/

Table 10: Diagnosis decision-making performance of LLMs. We use zero-shot prompting for rows 1-18, and we fine-tune the model on diagnosis decision training data for row 19. We report precision, recall, and F1 scores while using different code abstraction levels. The "[$n$]" notation following the model name indicated that the model is fine-tuned from the model in row $n$. † indicates the model is trained on biomedical or clinical resources. **Bold** marks the best model in each model group and green background indicates the best overall model.

| # | Model/Method | Params | Pre. | Rec. | F1 | Pre. | Rec. | F1 | Pre. | Rec. | F1 |
|---|---|---|---|---|---|---|---|---|---|---|---|
| | | | | Chapter | | | Category group | | | Category | |
| 1 | Flan T5 XL | 2.85B | 69.01 | 10.22 | 17.80 | 42.18 | 4.14 | 7.54 | 26.27 | 2.18 | 4.02 |
| 2 | Mistral v0.3 | 7B | 45.66 | 22.83 | 30.44 | 13.77 | 6.17 | 8.52 | 5.28 | 2.16 | 3.07 |
| 3 | Mistral Instruct v0.1 | 7B | 50.04 | 38.66 | 43.62 | 21.67 | 16.89 | 18.98 | 11.74 | 8.83 | 10.08 |
| 4 | Mistral Instruct v0.2 | 7B | 63.44 | 61.52 | 62.46 | 40.50 | 38.37 | 39.41 | 26.45 | 24.14 | 25.24 |
| 5 | Mistral Instruct v0.3 | 7B | 61.43 | **63.81** | 62.59 | 39.46 | 41.39 | 40.40 | 26.15 | 26.75 | 26.44 |
| 6 | BioMistral DARE† [3] | 7B | 56.33 | 28.06 | 37.46 | 29.05 | 12.65 | 17.62 | 17.32 | 6.89 | 9.85 |
| 7 | Mixtral Instruct v0.1 | 46.7B | **68.26** | 61.44 | **64.67** | **49.02** | **42.03** | **45.26** | **36.40** | **29.64** | **32.67** |
| 8 | LLaMA2 | 7B | 42.38 | 9.13 | 15.03 | 12.23 | 1.98 | 3.41 | 4.31 | 0.62 | 1.09 |
| 9 | LLaMA2 Instruct | 7B | 61.99 | **50.11** | 55.42 | **34.50** | **27.10** | **30.36** | **21.25** | **16.35** | **18.48** |
| 10 | Meditron† [8] | 7B | 42.73 | 27.77 | 33.66 | 11.62 | 7.26 | 8.93 | 2.98 | 1.73 | 2.19 |
| 11 | Asclepius† [8] | 7B | **64.66** | 12.70 | 21.23 | 31.04 | 4.90 | 8.47 | 17.00 | 2.53 | 4.40 |
| 12 | LLaMA3 | 8B | 51.03 | 34.86 | 41.42 | 21.33 | 16.17 | 18.39 | 11.70 | 9.23 | 10.32 |
| 13 | LLaMA3 Instruct | 8B | **63.00** | 57.82 | 60.30 | 39.26 | 34.61 | 36.79 | 27.09 | 22.53 | 24.60 |
| 14 | LLaMA3.1 Instruct | 8B | 59.09 | 68.99 | 63.66 | 35.04 | 44.48 | 39.20 | 23.68 | 30.03 | 26.48 |
| 15 | LLaMA3.2 Instruct | 1B | 58.03 | 52.61 | 55.19 | 30.19 | 27.87 | 28.98 | 17.58 | 16.12 | 16.82 |
| 16 | LLaMA3.2 Instruct | 3B | 59.40 | 63.24 | 61.26 | 31.72 | 36.42 | 33.90 | 19.43 | 22.48 | 20.85 |
| 17 | OpenBioLLM† [12] | 8B | 59.91 | 30.43 | 40.36 | 36.20 | 15.92 | 22.11 | 25.37 | 10.29 | 14.64 |
| 18 | LLaMA3 Instruct | 70B | 61.60 | **75.44** | **67.82** | **46.23** | **58.31** | **51.57** | **36.84** | **44.10** | **40.15** |
| 19 | GPT-3.5 turbo | — | **70.25** | 64.48 | 67.24 | **50.74** | 45.83 | 48.16 | **38.50** | 33.55 | 35.86 |
| 20 | GPT-4 turbo | — | 67.29 | 73.72 | 70.36 | 47.29 | 51.47 | 49.29 | 36.71 | 37.81 | 37.25 |
| 21 | GPT-4o | — | 66.70 | **80.96** | **73.15** | 48.73 | **63.99** | **55.33** | 36.90 | **48.64** | **41.97** |
| 22 | SFT [13] | 8B | 61.17 | 58.87 | 60.00 | 40.21 | 37.14 | 38.61 | 28.36 | 25.45 | 26.83 |

| # | Model/Method | Params | Pre. | Rec. | F1 | Pre. | Rec. | F1 | Pre. | Rec. | F1 |
|---|---|---|---|---|---|---|---|---|---|---|---|
| | | | | Sub-category | | | Full code | | | Average | |
| 1 | Flan T5 XL | 2.85B | 13.04 | 1.02 | 1.89 | 7.49 | 0.57 | 1.07 | 31.60 | 3.63 | 6.46 |
| 2 | Mistral v0.3 | 7B | 1.88 | 0.75 | 1.07 | 0.90 | 0.36 | 0.51 | 13.50 | 6.45 | 8.72 |
| 3 | Mistral Instruct v0.1 | 7B | 4.93 | 3.71 | 4.24 | 3.13 | 2.35 | 2.69 | 18.30 | 14.09 | 15.92 |
| 4 | Mistral Instruct v0.2 | 7B | 11.90 | 11.09 | 11.48 | 8.50 | 7.87 | 8.17 | 30.16 | 28.60 | 29.35 |
| 5 | Mistral Instruct v0.3 | 7B | 12.87 | 13.20 | 13.03 | 9.44 | 9.61 | 9.53 | 29.87 | 30.95 | 30.40 |
| 6 | BioMistral DARE† [3] | 7B | 7.75 | 3.05 | 4.37 | 4.87 | 1.91 | 2.74 | 23.06 | 10.51 | 14.41 |
| 7 | Mixtral Instruct v0.1 | 46.7B | **19.88** | **16.07** | **17.77** | **14.87** | **11.91** | **13.23** | **37.69** | **32.22** | **34.72** |
| 8 | LLaMA2 | 7B | 2.39 | 0.33 | 0.58 | 2.05 | 0.29 | 0.51 | 12.67 | 2.47 | 4.12 |
| 9 | LLaMA2 Instruct | 7B | **8.92** | **6.82** | **7.73** | **5.84** | **4.44** | **5.04** | **26.50** | **20.96** | **23.40** |
| 10 | Meditron† [8] | 7B | 0.91 | 0.52 | 0.66 | 0.55 | 0.31 | 0.40 | 11.76 | 7.52 | 9.17 |
| 11 | Asclepius† [8] | 7B | 7.04 | 1.04 | 1.81 | 3.91 | 0.58 | 1.01 | 24.73 | 4.35 | 7.38 |
| 12 | LLaMA3 | 8B | 5.00 | 4.48 | 4.73 | 3.35 | 3.14 | 3.24 | 18.48 | 13.58 | 15.62 |
| 13 | LLaMA3 Instruct | 8B | 12.94 | 10.80 | 11.78 | 9.24 | 7.68 | 8.39 | 30.31 | 26.69 | 28.37 |
| 14 | LLaMA3.1 Instruct | 8B | 12.14 | 15.75 | 13.71 | 8.85 | 11.42 | 9.97 | 27.76 | 34.14 | 30.61 |
| 15 | LLaMA3.2 Instruct | 1B | 8.26 | 7.66 | 7.95 | 5.54 | 5.15 | 5.34 | 23.92 | 21.88 | 22.86 |
| 16 | LLaMA3.2 Instruct | 3B | 9.35 | 11.08 | 10.14 | 6.26 | 7.42 | 6.79 | 25.23 | 28.13 | 26.59 |
| 17 | OpenBioLLM† [12] | 8B | 12.69 | 4.97 | 7.15 | 8.28 | 3.21 | 4.63 | 28.49 | 12.97 | 17.78 |
| 18 | LLaMA3 Instruct | 70B | **23.53** | **27.58** | **25.39** | **18.76** | **21.90** | **20.21** | **37.39** | **45.46** | **41.03** |
| 19 | GPT-3.5 turbo | — | 25.81 | 22.36 | 23.96 | 21.52 | 18.61 | 19.96 | **41.36** | 36.96 | 39.03 |
| 20 | GPT-4 turbo | — | 22.29 | 22.42 | 22.35 | 17.26 | 17.18 | 17.22 | 38.17 | 40.52 | 39.30 |
| 21 | GPT-4o | — | **28.26** | **37.12** | **32.09** | **24.31** | **31.86** | **27.58** | 40.98 | **52.52** | **46.02** |
| 22 | SFT [13] | 8B | 13.42 | 12.17 | 12.76 | 9.72 | 8.87 | 9.27 | 30.58 | 28.50 | 29.50 |

Table 11: Procedure decision-making performance of LLMs.

| # | Model | Param # | Pre. | Rec. Level 1 | F1 | Pre. | Rec. Level 2 | F1 | Pre. | Rec. Level 3 | F1 |
|---|---|---|---|---|---|---|---|---|---|---|---|
| 1 | Flan T5 XL | 2.85B | 10.81 | 7.03 | 8.52 | 1.39 | 0.73 | 0.96 | 0.37 | 0.17 | 0.24 |
| 2 | Mistral v0.3 | 7B | 19.98 | 24.81 | 22.14 | 2.83 | 3.33 | 3.06 | 0.73 | 0.81 | 0.77 |
| 3 | Mistral Instruct v0.1 | 7B | 21.32 | 48.89 | **29.69** | 9.90 | 23.01 | 13.85 | 3.40 | 7.81 | 4.73 |
| 4 | Mistral Instruct v0.2 | 7B | **21.57** | 44.71 | 29.10 | **11.21** | 21.98 | 14.85 | **4.34** | 8.13 | 5.66 |
| 5 | Mistral Instruct v0.3 | 7B | 19.65 | **50.92** | 28.36 | 10.15 | **24.99** | 14.43 | 3.59 | 8.77 | 5.09 |
| 6 | BioMistral DARE† [3] | 7B | 21.29 | 27.80 | 24.12 | 10.21 | 12.28 | 11.15 | 2.94 | 3.28 | 3.10 |
| 7 | Mistral Instruct v0.1 | 46.7B | 19.82 | 48.49 | 28.14 | 10.74 | 24.61 | **14.95** | 4.20 | **9.37** | 5.80 |
| 8 | LLaMA2 | 7B | 11.53 | 11.36 | 11.45 | 3.54 | 3.33 | 3.43 | 0.71 | 0.65 | 0.68 |
| 9 | LLaMA2 instruct | 7B | 19.56 | **35.32** | 25.18 | 10.05 | **16.28** | **12.43** | 3.28 | **5.13** | 4.00 |
| 10 | Meditron† [8] | 8B | 19.13 | 21.32 | 20.17 | 2.57 | 2.52 | 2.55 | 0.65 | 0.59 | 0.62 |
| 11 | Asclepius† [8] | 8B | **25.93** | 18.29 | 21.45 | **14.78** | 8.73 | 10.98 | **5.98** | 3.21 | **4.18** |
| 12 | LLaMA3 | 8B | 20.74 | 35.52 | 26.19 | 5.19 | 9.42 | 6.69 | 1.58 | 2.93 | 2.06 |
| 13 | LLaMA3 Instruct | 8B | 20.95 | 56.41 | 30.56 | 9.52 | 27.04 | 14.09 | **3.89** | 10.86 | **5.73** |
| 14 | LLaMA3.1 Instruct | 8B | 18.72 | **64.09** | 28.98 | 8.29 | **31.82** | 13.15 | 3.30 | **12.64** | 5.24 |
| 15 | LLaMA3.2 Instruct | 1B | 17.35 | 50.21 | 25.79 | 7.71 | 23.17 | 11.57 | 2.85 | 8.43 | 4.26 |
| 16 | LLaMA3.2 Instruct | 3B | 18.21 | 60.48 | 27.99 | 9.31 | 31.53 | **14.37** | 3.32 | 11.32 | 5.14 |
| 17 | OpenBioLLM† [12] | 8B | **22.28** | 34.50 | 27.08 | **9.61** | 13.55 | 11.25 | 3.54 | 4.69 | 4.03 |
| 18 | LLaMA3 Instruct | 70B | 20.86 | 58.59 | **30.77** | 8.82 | 26.42 | 13.23 | 3.73 | 10.86 | 5.56 |
| 19 | GPT-3.5 turbo | — | **21.14** | 55.40 | **30.60** | 11.22 | 28.95 | **16.17** | 4.13 | 10.47 | 5.92 |
| 20 | GPT-4 turbo | — | 18.47 | 57.06 | 27.91 | 10.01 | 29.19 | 14.91 | 3.84 | 11.37 | 5.74 |
| 21 | GPT-4o | — | 19.54 | **62.79** | 29.80 | 9.91 | **32.54** | 15.19 | **4.54** | **14.69** | **6.94** |

| # | Model | Param # | Pre. | Rec. Full code | F1 | Pre. | Rec. Average | F1 |
|---|---|---|---|---|---|---|---|---|
| 1 | Flan T5 XL | 2.85B | 0.00 | 0.00 | 0.00 | 3.14 | 1.98 | 2.43 |
| 2 | Mistral v0.3 | 7B | 0.00 | 0.00 | 0.00 | 5.89 | 7.24 | 6.49 |
| 3 | Mistral Instruct v0.1 | 7B | 0.17 | 0.76 | 0.28 | 8.70 | 20.12 | 12.14 |
| 4 | Mistral Instruct v0.2 | 7B | 0.27 | 0.88 | 0.41 | **9.35** | 18.93 | **12.51** |
| 5 | Mistral Instruct v0.3 | 7B | 0.26 | **1.11** | **0.42** | 8.41 | **21.45** | 12.07 |
| 6 | BioMistral DARE† [3] | 7B | **0.30** | 0.48 | 0.37 | 8.68 | 10.96 | 9.68 |
| 7 | Mixtral Instruct v0.1 | 46.7B | 0.20 | 0.80 | 0.32 | 8.74 | 20.82 | 12.30 |
| 8 | LLaMA2 | 7B | 0.00 | 0.00 | 0.00 | 3.95 | 3.84 | 3.89 |
| 9 | LLaMA2 Instruct | 7B | 0.19 | **0.50** | 0.27 | 8.27 | **14.31** | **10.47** |
| 10 | Meditron† [8] | 7B | 0.04 | 0.04 | 0.04 | 5.60 | 6.12 | 5.84 |
| 11 | Asclepius† [8] | 7B | **0.86** | 0.46 | **0.60** | **11.89** | 7.67 | 9.30 |
| 12 | LLaMA3 | 8B | **0.28** | 1.46 | **0.47** | 8.66 | 23.94 | 12.71 |
| 13 | LLaMA3 Instruct | 8B | 0.23 | 1.19 | 0.39 | **8.80** | 24.08 | **12.88** |
| 14 | LLaMA3.1 Instruct | 8B | 0.22 | **1.61** | 0.39 | 7.63 | **27.54** | 11.94 |
| 15 | LLaMA3.2 Instruct | 1B | 0.15 | 0.77 | 0.26 | 7.02 | 20.64 | 10.47 |
| 16 | LLaMA3.2 Instruct | 3B | 0.14 | 0.88 | 0.24 | 7.74 | 26.06 | 11.93 |
| 17 | OpenBioLLM† [12] | 8B | 0.08 | 0.25 | 0.12 | 6.90 | 12.03 | 8.76 |
| 18 | LLaMA3 Instruct | 70B | 0.20 | 1.19 | 0.35 | 8.41 | 24.26 | 12.48 |
| 19 | GPT-3.5 turbo | — | 0.27 | 1.19 | 0.44 | **9.19** | 24.00 | 13.28 |
| 20 | GPT-4 turbo | — | 0.19 | 1.00 | 0.33 | 8.13 | 24.65 | 12.22 |
| 21 | GPT-4o | — | **0.39** | **2.30** | **0.67** | 8.59 | **28.08** | 13.15 |

Table 12: Lab tests ordering performance of LLMs.

| # | Model | Param # | Pre. | Rec. Level 1 | F1 | Pre. | Rec. Level 2 | F1 | Pre. | Rec. Level 3 | F1 |
|---|---|---|---|---|---|---|---|---|---|---|---|
| 1 | Flan T5 XL | 2.85B | 100.00 | 99.81 | 99.91 | 95.02 | 42.28 | 58.52 | 78.85 | 8.43 | 15.24 |
| 2 | Mistral v0.3 | 7B | 99.54 | 99.69 | 99.62 | 80.02 | 56.44 | 66.19 | 51.93 | 18.17 | 26.92 |
| 3 | Mistral Instruct v0.1 | 7B | 99.62 | 99.81 | 99.71 | 77.29 | 74.14 | 75.68 | 55.87 | 40.41 | 46.90 |
| 4 | Mistral Instruct v0.2 | 7B | 99.72 | 99.81 | 99.77 | 76.07 | 76.33 | 76.20 | 57.12 | 38.79 | 46.21 |
| 5 | Mistral Instruct v0.3 | 7B | **99.81** | 99.81 | **99.81** | 77.15 | **78.38** | **77.76** | 53.73 | **43.96** | **48.36** |
| 6 | BioMistral DARE[†] [3] | 7B | 99.56 | **99.85** | 99.70 | **81.71** | 63.99 | 71.77 | **64.87** | 28.62 | 39.72 |
| 7 | Mixtral Instruct v0.1 | 46.7B | **99.81** | 99.81 | **99.81** | 77.85 | 75.41 | 76.61 | 53.76 | 38.92 | 45.16 |
| 8 | LLaMA2 | 7B | 99.56 | 99.78 | 99.67 | **90.37** | 46.32 | 61.25 | 55.13 | 11.87 | 19.54 |
| 9 | LLaMA2 Instruct | 7B | **100.00** | 99.89 | **99.95** | 75.55 | **64.56** | **69.62** | 58.38 | **33.65** | **42.69** |
| 10 | Meditron[†] [8] | 7B | 99.90 | 99.79 | 99.84 | 81.34 | 52.49 | 63.81 | **58.59** | 13.46 | 21.90 |
| 11 | Asclepius[†] [8] | 7B | 99.80 | 99.80 | 99.80 | 84.88 | 43.83 | 57.81 | 56.77 | 12.18 | 20.06 |
| 12 | LLaMA3 | 8B | 97.23 | 99.80 | 98.50 | 73.77 | 73.03 | 73.40 | 39.07 | 36.52 | 37.75 |
| 13 | LLaMA3 Instruct | 8B | 99.91 | **99.81** | 99.86 | 75.69 | 78.67 | 77.15 | 51.09 | 47.04 | 48.98 |
| 14 | LLaMA3.1 Instruct | 8B | **100.00** | 99.81 | **99.91** | 74.10 | 81.01 | 77.40 | 47.82 | 49.56 | 48.67 |
| 15 | LLaMA3.2 Instruct | 1B | 99.91 | **99.81** | 99.86 | 75.63 | 77.25 | 76.43 | 49.57 | 40.89 | 44.82 |
| 16 | LLaMA3.2 Instruct | 3B | **100.00** | 99.81 | **99.91** | 73.62 | **81.14** | 77.20 | 46.80 | **50.08** | 48.38 |
| 17 | OpenBioLLM[†] [12] | 8B | 99.81 | 99.62 | 99.72 | **78.97** | 67.69 | 72.89 | **53.56** | 28.42 | 37.14 |
| 18 | LLaMA3 Instruct | 70B | **100.00** | 99.81 | **99.91** | 74.97 | 80.18 | **77.49** | 53.12 | 45.63 | **49.09** |
| 19 | GPT-3.5 turbo | — | **100.00** | 99.81 | **99.91** | 79.36 | 71.89 | 75.44 | **59.34** | 41.43 | **48.79** |
| 20 | GPT-4 turbo | — | 99.91 | **99.81** | 99.86 | 77.91 | 73.57 | 75.67 | 54.12 | 44.20 | 48.66 |
| 21 | GPT-4o | — | 99.91 | **99.81** | 99.86 | 77.93 | 74.86 | 76.37 | 53.02 | **44.85** | 48.60 |

| # | Model | Param # | Pre. | Rec. Level 4 | F1 | Pre. | Rec. Average | F1 |
|---|---|---|---|---|---|---|---|---|
| 1 | Flan T5 XL | 2.85B | 49.72 | 1.86 | 3.58 | 80.90 | 38.10 | 44.31 |
| 2 | Mistral v0.3 | 7B | 34.93 | 4.50 | 7.97 | 66.61 | 44.70 | 50.18 |
| 3 | Mistral Instruct v0.1 | 7B | 31.71 | 10.68 | 15.97 | 66.12 | 56.26 | 59.57 |
| 4 | Mistral Instruct v0.2 | 7B | 33.39 | 9.71 | 15.04 | 66.57 | 56.16 | 59.30 |
| 5 | Mistral Instruct v0.3 | 7B | 30.42 | **11.48** | **16.67** | 65.28 | **58.41** | 60.65 |
| 6 | BioMistral DARE[†] [3] | 7B | **35.59** | 7.00 | 11.70 | **70.43** | 49.86 | 55.72 |
| 7 | Mixtral Instruct v0.1 | 46.7B | 31.56 | 9.37 | 14.45 | 65.75 | 55.88 | 59.01 |
| 8 | LLaMA2 | 7B | **57.56** | 3.81 | 7.15 | **75.66** | 40.45 | 46.90 |
| 9 | LLaMA2 Instruct | 7B | 24.93 | **7.06** | **11.01** | 64.71 | **51.29** | 55.82 |
| 10 | Meditron[†] [8] | 7B | 41.12 | 3.26 | 6.04 | 70.24 | 42.25 | 47.90 |
| 11 | Asclepius[†] [8] | 7B | 26.12 | 3.08 | 5.52 | 66.90 | 39.73 | 45.80 |
| 12 | LLaMA3 | 8B | 26.30 | 11.53 | 16.03 | 59.09 | 55.22 | 56.42 |
| 13 | LLaMA3 Instruct | 8B | 29.98 | 13.68 | 18.78 | 64.17 | 59.80 | 61.19 |
| 14 | LLaMA3.1 Instruct | 8B | 28.26 | 15.31 | 19.86 | 62.55 | 61.42 | 61.46 |
| 15 | LLaMA3.2 Instruct | 1B | 31.34 | 10.65 | 15.90 | 64.11 | 57.15 | 59.25 |
| 16 | LLaMA3.2 Instruct | 3B | 32.33 | **16.48** | **21.83** | 63.19 | **61.88** | **61.83** |
| 17 | OpenBioLLM[†] [12] | 8B | **37.97** | 6.92 | 11.71 | **67.58** | 50.66 | 55.36 |
| 18 | LLaMA3 Instruct | 70B | 26.02 | 10.60 | 15.07 | 63.53 | 59.05 | 60.39 |
| 19 | GPT-3.5 turbo | — | **33.69** | 11.76 | **17.43** | **68.10** | 56.22 | 60.39 |
| 20 | GPT-4 turbo | — | 27.58 | 10.58 | 15.29 | 64.88 | 57.04 | 59.87 |
| 21 | GPT-4o | — | 27.26 | **12.54** | 17.18 | 64.53 | **58.02** | **60.50** |

Table 13: Prescriptions decision-making performance of LLMs.

| # | Model | Param # | Pre. | Rec. Level 1 | F1 | Pre. | Rec. Level 2 | F1 | Pre. | Rec. Level 3 | F1 |
|---|---|---|---|---|---|---|---|---|---|---|---|
| 1 | Flan T5 XL | 2.85B | 76.75 | 13.73 | 23.30 | 73.45 | 11.44 | 19.79 | 42.51 | 4.41 | 7.99 |
| 2 | Mistral v0.3 | 7B | 75.29 | 33.11 | 45.99 | 70.44 | 28.84 | 40.92 | 38.04 | 12.83 | 19.19 |
| 3 | Mistral Instruct v0.1 | 7B | 77.16 | 59.09 | 66.93 | 74.24 | 54.18 | 62.65 | 44.46 | 29.81 | 35.69 |
| 4 | Mistral Instruct v0.2 | 7B | 84.09 | 56.70 | 67.73 | 81.89 | 53.03 | 64.38 | 54.33 | 31.84 | 40.15 |
| 5 | Mistral Instruct v0.3 | 7B | 80.30 | **68.47** | 73.91 | 78.01 | **64.21** | 70.44 | 50.35 | 40.53 | 44.91 |
| 6 | BioMistral DARE[†] [3] | 7B | 81.47 | 39.64 | 53.33 | 78.17 | 35.40 | 48.73 | 48.31 | 17.58 | 25.78 |
| 7 | Mixtral Instruct v0.1 | 46.7B | **84.52** | 66.17 | **74.23** | **82.23** | 63.32 | **71.55** | **57.80** | 41.98 | **48.64** |
| 8 | LLaMA2 | 7B | 79.88 | 23.74 | 36.60 | 76.04 | 20.54 | 32.35 | 34.63 | 7.59 | 12.45 |
| 9 | LLaMA2 Instruct | 7B | 78.78 | **52.94** | **63.32** | 75.87 | **48.74** | **59.35** | 49.19 | **29.80** | **37.11** |
| 10 | Meditron[†] [8] | 7B | 74.64 | 26.65 | 39.27 | 69.51 | 22.45 | 33.94 | 34.22 | 8.05 | 13.03 |
| 11 | Asclepius[†] [8] | 7B | **86.32** | 17.14 | 28.60 | **84.27** | 14.73 | 25.08 | **50.70** | 6.60 | 11.68 |
| 12 | LLaMA3 | 8B | 76.22 | 54.90 | 63.82 | 72.85 | 50.72 | 59.81 | 41.68 | 31.00 | 35.55 |
| 13 | LLaMA3 Instruct | 8B | 83.03 | 69.10 | 75.43 | 80.94 | 65.82 | 72.60 | 54.94 | 43.44 | 48.52 |
| 14 | LLaMA3.1 Instruct | 8B | 79.37 | **79.26** | **79.32** | 77.19 | **76.30** | **76.74** | 48.97 | **52.13** | 50.50 |
| 15 | LLaMA3.2 Instruct | 1B | 79.16 | 68.31 | 73.34 | 77.30 | 63.59 | 69.78 | 48.16 | 39.56 | 43.44 |
| 16 | LLaMA3.2 Instruct | 3B | 77.24 | 74.89 | 76.05 | 75.12 | 70.94 | 72.97 | 44.57 | 45.65 | 45.10 |
| 17 | OpenBioLLM[†] [12] | 8B | **84.89** | 45.42 | 59.18 | **81.99** | 41.21 | 54.85 | 56.89 | 24.11 | 33.87 |
| 18 | LLaMA3 Instruct | 70B | 83.20 | 73.54 | 78.07 | 81.48 | 70.26 | 75.45 | **58.09** | 49.74 | **53.59** |
| 19 | GPT-3.5 turbo | — | 85.26 | 57.30 | 68.54 | 82.85 | 52.87 | 64.55 | 58.69 | 33.26 | 42.46 |
| 20 | GPT-4 turbo | — | **86.13** | 66.66 | 75.15 | **84.28** | 63.95 | 72.72 | **64.38** | 45.40 | 53.25 |
| 21 | GPT-4o | — | 84.93 | **70.58** | **77.09** | 83.05 | **67.46** | 74.45 | 62.58 | **49.43** | **55.24** |

| # | Model | Param # | Pre. | Rec. Level 4 | F1 | Pre. | Rec. Average | F1 |
|---|---|---|---|---|---|---|---|---|
| 1 | Flan T5 XL | 2.85B | 27.56 | 2.42 | 4.45 | 55.07 | 8.00 | 13.88 |
| 2 | Mistral v0.3 | 7B | 27.28 | 8.50 | 12.96 | 52.76 | 20.82 | 29.76 |
| 3 | Mistral Instruct v0.1 | 7B | 33.64 | 21.54 | 26.27 | 57.38 | 41.16 | 47.88 |
| 4 | Mistral Instruct v0.2 | 7B | 45.25 | 24.30 | 31.62 | 66.39 | 41.47 | 50.97 |
| 5 | Mistral Instruct v0.3 | 7B | 40.94 | 30.87 | 35.20 | 62.40 | 51.02 | 56.11 |
| 6 | BioMistral DARE[†] [3] | 7B | 38.73 | 12.77 | 19.21, | 61.67 | 26.35 | 36.76 |
| 7 | Mixtral Instruct v0.1 | 46.7B | **49.14** | **32.96** | **39.46** | **68.42** | 51.11 | **58.47** |
| 8 | LLaMA2 | 7B | 26.21 | 5.33 | 8.86 | 54.19 | 14.30 | 22.56 |
| 9 | LLaMA2 Instruct | 7B | **40.38** | **23.30** | **29.55** | 61.06 | **38.69** | **47.33** |
| 10 | Meditron[†] [8] | 7B | 24.60 | 5.11 | 8.46, | 50.74 | 15.56 | 23.68 |
| 11 | Asclepius[†] [8] | 7B | 39.60 | 4.81 | 8.57 | **65.22** | 10.82 | 18.49 |
| 12 | LLaMA3 | 8B | 32.58 | 23.92 | 27.59 | 55.83 | 40.13 | 46.69 |
| 13 | LLaMA3 Instruct | 8B | 46.43 | 34.57 | 39.63 | 66.33 | 53.23 | 59.04 |
| 14 | LLaMA3.1 Instruct | 8B | 39.63 | **40.90** | 40.26 | 61.29 | **62.15** | 61.70 |
| 15 | LLaMA3.2 Instruct | 1B | 38.50 | 30.43 | 33.99 | 60.78 | 50.47 | 55.14 |
| 16 | LLaMA3.2 Instruct | 3B | 34.18 | 34.21 | 34.20 | 57.78 | 56.42 | 57.08 |
| 17 | OpenBioLLM[†] [12] | 8B | 48.18 | 18.76 | 27.00 | 67.99 | 32.38 | 43.72 |
| 18 | LLaMA3 Instruct | 70B | **50.82** | 40.74 | **45.22** | **68.40** | 58.57 | 63.08 |
| 19 | GPT-3.5 turbo | — | 50.38 | 26.51 | 34.74 | 69.30 | 42.48 | 52.57 |
| 20 | GPT-4 turbo | — | **58.00** | 37.83 | 45.79 | **73.20** | 53.46 | 61.73 |
| 21 | GPT-4o | — | 55.45 | **41.42** | **47.42** | 71.50 | **57.22** | **63.55** |