# OpenReview forum: "CliBench: A Multifaceted and Multigranular Evaluation of Large Language Models for Clinical Decision Making"
_ICLR.cc/2025/Conference — Submitted to ICLR 2025_

### Official Review · Reviewer_P3hN · 2024-11-02

**Soundness:** 2
**Presentation:** 3
**Contribution:** 1
**Rating:** 3
**Confidence:** 3

**Summary:**

The paper introduces CLIBENCH, a new benchmark for evaluating Large Language Models in clinical decision-making. CLIBENCH utilizes the MIMIC-IV dataset to offer a multi-faceted evaluation across various specialties and clinical tasks, including diagnosis, procedure identification, lab test ordering, and medication prescription. CLIBENCH employs structured ontologies (ICD-10-CM, ICD-10-PCS, LOINC, ATC) to enable precise, multi-granular evaluation. The authors conduct a zero-shot evaluation of several LLMs, highlighting their potential and limitations in clinical settings.

**Strengths:**

The use of ontologies and hierarchical evaluation provides a more nuanced understanding of LLM capabilities at different levels of granularity.

The inclusion of multiple clinical tasks across various specialties makes the open-source benchmark more representative of real-world clinical practice.

**Weaknesses:**

The use of publicly available MIMIC-IV data raises concerns about potential data leakage during LLM pre-training. While the authors discuss this, they should consider strengthening their argument by conducting further analysis or experiments to quantify the impact of potential leakage. In addition, solely on MIMIC-IV from a single medical center limits the generalizability of findings.

The absence of few-shot/ICL experiments restricts the understanding of LLM adaptability and learning potential in these tasks. Zero-shot setting on either clinical reasoning or outputting desired ontologies/structured outputs is not a realistic task.

While the authors justify using billing codes as a proxy for "collective knowledge," including a physician performance baseline would provide a more direct comparison and better contextualize LLM performance.

Ground truth relies solely on EHR records (line 198). The authors do not mention obtaining physician agreements or validation of the ground truth labels used in CLIBENCH.

**Questions:**

How sensitive are the results to variations in prompt phrasing? Did the authors explore different prompt variations?

Could the authors elaborate on the limitations of using billing codes as a proxy for ground truth diagnoses?

How might CLIBENCH be extended to incorporate temporal aspects of clinical decision-making? As the patients are not really coming in with all the notes.

---

### Official Review · Reviewer_h2gq · 2024-11-03

**Soundness:** 2
**Presentation:** 3
**Contribution:** 3
**Rating:** 5
**Confidence:** 4

**Summary:**

This paper presents CliBench, a benchmark including four important clinical tasks - diagnosis, treatment procedures, lab test orders, and prescriptions, developed from the publicly available MIMIC IV dataset. This paper conducts a comprehensive evaluation of the zero-shot performance of leading LLMs, including both open-source and proprietary models, general domain and medical specific models, on these four tasks. The findings highlight both the potential and the limitations of current LLMs in real-world clinical decision support.

**Strengths:**

1. This paper is clearly written and easy to follow.
2. The selected four tasks are both common and important in real-world clinical settings. A benchmark including these tasks provides researchers in this field with a more comprehensive and practical way to evaluate LLMs for real-world clinical applications.
3. This paper compares the zero-shot performance of a wide range of LLMs, and evaluates them at different levels of diagnostic, procedure, lab test, and prescription codes. This approach enables a thorough evaluation of the capabilities of LLMs in clinical decision-making across varying degrees of resolution, offering a detailed understanding of their strengths and weaknesses.
4. The insights gained from analyzing the zero-shot performance of various leading LLMs are very interesting, and many shed light on potential future research directions such as post-training and instruction tuning.

**Weaknesses:**

1. For procedures, lab tests, and prescriptions, the inputs are patient profiles and medical records. Is this information sufficient for human clinical experts to make real-world decisions? If clinicians need to rely on additional data, such as ICU monitoring, it might be unrealistic to expect LLMs to make accurate decisions solely based on such information. Section 6.3 shows that missing important patient information can significantly impact LLM performance. If this subset of information aligns with what clinicians use, it would be helpful to justify the design and the validity of the benchmark in the paper.

2. Section 3.3 mentions that admissions without records for the target tasks are filtered out. For AI systems deployed for decision support, I think it is also important for them to recognize cases where no procedure, lab test, or prescription is required (i.e., true negatives). Is there a plan to include such "negative" admissions in the evaluation set, or a justification that they should not be included?

3. In section 3.6, micro-level precision, recall, and F1 are used as metrics. Figure 1(b) shows significant variance in the number of cases across disease types. Including macro-level metrics or disease type-specific metrics (similar to the patient breakdown analysis in Section 6.2) would provide more insights into the strengths and weaknesses of LLMs across different disease types, e.g. some LLMs might perform well on certain disease types but poorly on others, while other LLMs might demonstrate different behavior.

Note: I am not a clinician and may not fully understand certain aspects of clinical decision-making. I am open to changing my score based on clarifications regarding these concerns.

**Questions:**

1. In line 209 - 214, the sampling are conducted on different levels for different targets, e.g. top-level for procedure codes and third-level for lab tests. What is the consideration behind different levels? It might be helpful to mention it clearly in the paper.
2. In line 237 - 239, is the BERT used general domain or biomedical specific? If it is a general-domain paper, can the sentence embeddings accurately capture medical specific terms and determine the similarity between the LLM responses and code descriptions?
3. In line 338 - 340, is the LLaMA3 SFT fine-tuned by next-token prediction on ground-truth diagnosis, procedures etc.? For admissions with multiple diagnosis (or other tasks), how would the order of multiple labels in the output impact the SFT performance? Did you do any permutation or data processing to prevent potential bias?
4. In section 5, it might be helpful to use the actual model names rather than row numbers when referring to models, so that readers would not have to refer back to Table 3 to understand the performance differences between different models.
5. In Table 4, it is very interesting that all models achieve 99% F1 for Level 1, and also high F1 for Level 2. Is it because there are only 2/5 candidates for lab test order level 1/2, so that models can achieve good performance easily? I think it would be helpful if you could provide one or two sentence explanation for this so that it gives readers more intuitive insights on this performance.

---

### Official Review · Reviewer_Tkst · 2024-11-04

**Soundness:** 2
**Presentation:** 3
**Contribution:** 2
**Rating:** 6
**Confidence:** 4

**Summary:**

This paper introduces CLIBENCH, a comprehensive benchmark for evaluating large language models  in clinical decision-making, designed to address limitations of prior clinical evaluations. Unlike previous benchmarks that focus narrowly on single diseases or specialties, CLIBENCH provides a broad, multi-specialty assessment across key clinical tasks: diagnoses, procedure recommendations, lab test ordering, and prescriptions. Built on the MIMIC-IV dataset, CLIBENCH employs structured medical ontologies to support multi-granular evaluation, enabling both coarse and fine-grained assessments. The authors evaluate a range of LLMs, including general-purpose, instruction-tuned, and domain-specific models, in a zero-shot setting, revealing strengths and limitations of current models in clinical contexts and highlighting areas for further development in LLM-powered healthcare.

**Strengths:**

The submission introduces CLIBENCH, a well-structured benchmark that addresses clinical decision-making, covering diagnoses, procedures, lab test orders, and prescriptions. This benchmark extends beyond typical disease-focused evaluations​

1. Real-World Clinical Relevance: By leveraging MIMIC-IV dataset and aligning tasks with real-world clinical workflows, such as lab test ordering and treatment procedure identification, CLIBENCH captures the complexity of real clinical settings, which bridge a benchmark's relevance to practical applications​

2. Metrics:  detailed, multi-granular evaluation metric across various clinical tasks, at different abstraction levels, such as chapter and category for ICD-10 codes​

3. Evaluation: a comprehensive evaluation of multiple open-source and closed-source LLMs, as well as one finetuned model, presenting clear results on the performance gaps

**Weaknesses:**

- Reliance on Zero-Shot Evaluations: Although the benchmark provides valuable insights, the study primarily relies on zero-shot evaluations, despite the sensitivity of LLM performance to prompt configurations. Exploring few-shot settings or incorporating multi-prompt variations could enhance robustness and offer a more comprehensive assessment of the benchmark's applicability.

- "Domain-specialized models do not work": This statement could benefit from a more nuanced analysis, with additional experiments to substantiate the findings. A deeper exploration into why domain-specific adaptations underperform in this context would provide valuable insights and help clarify the challenges involved.

- Lower Performance on Specific Tasks: Performance on several tasks, particularly procedures and lab test orders, is noticeably lower. Further investigation would clarify whether this discrepancy reflects inherent model limitations or potential areas to strengthen the benchmark's design.

- Ambiguity in Prompt Construction: While prompt construction is outlined, the study lacks an in-depth evaluation of prompt effectiveness across different clinical tasks. Analyzing how specific prompt structures influence model performance could yield valuable insights and improve task-specific optimization.

**Questions:**

- Domain-Specific Model Performance: The claim that domain-specialized models don’t improve performance needs clarification. Could this underperformance be due to model training limitations, data biases, or potential improvements needed in benchmark design?

- Zero-Shot Evaluation: Given LLMs' sensitivity to prompts, would the authors consider testing few-shot settings or experimenting with prompt variations to assess models robustness?

---

### Official Review · Reviewer_YzcX · 2024-11-04

**Soundness:** 1
**Presentation:** 2
**Contribution:** 2
**Rating:** 3
**Confidence:** 4

**Summary:**

This work introduces a new LLM benchmark to assess LLMs' capabilities in clinical diagnosis using the MIMIC-IV database. This benchmark primarily covers four tasks: predicting discharge diagnoses, procedures, lab test orders, and prescriptions. Since entries in MIMIC-IV are stored in a structured format with expert ontology, this enables multi-granular evaluation from coarse to fine levels. For experimental studies, this work presents the zero-shot performance of various LLMs (Section 5), along with additional analysis (Section 6).

**Strengths:**

- The new large-scale LLM evaluation benchmark in the medical domain could be highly beneficial for the research community.
- A variety of LLMs were evaluated in this paper, allowing researchers to observe trends and understand which models are at least operational within the MIMIC-IV dataset and the four clinical tasks suggested in this work.

**Weaknesses:**

- The main issue with this work is that the task formulation does not closely align with real-world clinical decision-making. Although the paper emphasizes that its tasks are grounded in clinical diagnosis (e.g., L61, Section 3.1) due to the use of the realistic MIMIC-IV dataset, there are significant aspects to consider, such as (1) the clinical decision-making process and (2) the benchmarking process itself.
- Regarding the connection of this benchmark to clinical decision-making, predicting ICD-10-CM diagnosis and ICD-10-PCS procedure codes is primarily a medical billing task, not a direct diagnosis or treatment task, which the authors have already recognized (see Section F. Limitations).
- Even if the benchmark assumes the ability to perform a billing task or a reasonable clinical workflow, its effectiveness as a benchmark is quite limited. Specifically, it is unclear whether sufficient input data has been properly formatted or extracted from MIMIC-IV for each task, raising doubts about whether a perfect score is achievable. Additionally, there is limited insight into how well LLMs are performing in this context. At a minimum, the paper should provide a baseline score, such as an expert clinician score or a majority-class prediction, to contextualize the LLMs' performance.

**Questions:**

- Beyond the seven insights from LLM results (bolded text in Section 5.1), what is the clinical significance of GPT-4o achieving a score of 27.58 in the full code for the diagnosis decision task? How far is this from clinician behavior?
- Similarly, in the statement "Models are less familiar with procedures and lab orders" (bolded text in Section 5.2), are clinicians familiar with procedures and lab orders at a fine-grained level such that they could achieve a 100% score?
- During the patient journey from admission to discharge, there is a significant amount of structured data in the MIMIC-IV database beyond clinical notes. Is it possible to predict these tasks without using all available patient data? If not 100%, what is the upper bound for each task?
- Additionally, there are truncated instances during benchmarking, which means that even if LLMs perform perfectly, some may not be able to achieve their potential high scores. I think the evaluation dataset instances should ideally fit within an 8K context length, or there should be enough LLMs capable of handling the full context length of the dataset.

---

### Official Review · Reviewer_N2Z1 · 2024-11-04

**Soundness:** 2
**Presentation:** 3
**Contribution:** 3
**Rating:** 5
**Confidence:** 4

**Summary:**

This paper presents a new benchmark CliBench developed from the MIMIC IV dataset, offering a comprehensive and realistic assessment of LLMs’ capabilities in clinical diagnosis. Specifically, the authors construct four clinical decision-making tasks, requiring LLMs to predict the clinical codes of diagnoses, procedures, lab tests, and prescriptions based on the information recorded in the EHR of a patient. The dataset is constructed through a rule-based NLP pipeline and verified by a clinical NLP expert. The author tested the performance of a range of LLMs on the constructed dataset, and the results show that current LLMs generally perform poorly on these clinical decision-making tasks, especially in procedure prediction. The author further analyzed the impact of patient attributes, task difficulties, and clinical data elements on diagnostic tasks.

**Strengths:**

+ The authors provide an open-source, multi-task evaluation set covering various clinical decision-making tasks, which will be highly beneficial for the development of future medical LLMs.

+ The authors conducted a systematic evaluation on a total of 20+ LLMs, including the powerful GPT-4o model.

+ The authors provide a detailed analysis of performance on diagnostic tasks, which may offer insights for the application of LLMs in diagnostics.

**Weaknesses:**

My main concern lies in the evaluation approach of this work. Unlike other mainstream medical benchmarks, the authors use clinical code prediction as the downstream task and employ precision, recall, and F1-score as performance metrics for the four tasks. While this approach is indeed closer to the real deployment environment of medical AI, it also introduces the following issues:

1.	**Test Prompt**: I noticed that the authors prompt the language model with “provide as many diagnoses as you can until you are not confident about your diagnosis decision.” Has the impact of different prompt styles on performance been tested? The current prompt format seems to encourage outputting as many predicted codes as possible, which may be a key reason why, in Figure 3c, the F1-score increases as the number of ground-truth diagnoses grows. I believe it is necessary to test prompts with different phrasing (e.g., “please provide an appropriately sized set of diagnoses”) to further improve the stability of the evaluation results.

2.	**Answer Extraction**: I noticed that the authors allow the language model to output either clinical codes or predictions in text form. For text-based predictions, they use a BERT model pretrained on 1B sentence pairs to calculate sentence similarity and select the closest code as the predicted result. I have the following questions:
	+ When the model provides both code and text-based results, is priority given to extracting the code or to parsing the text? In such cases, is the accuracy higher when parsing the code directly or when interpreting the text result?
	+ What is the accuracy of this text parsing method based on sentence embeddings, and has any related analysis been conducted? Are there alternative methods that could further improve matching accuracy?
3.	**Human Physician Performance**: Although the authors provide some reasons in the appendix for not including human physician performance, I still believe it is essential to add human performance data (even if on a small scale) for this dataset. First, the tasks in this dataset are inherently challenging; for example, ICD-10-CM contains over 70,000 codes, and ICD-10-PCS has over 87,000 codes. Even for medical experts, accurately completing the coding without consulting the specific ICD-10 code set is very difficult. Including evaluation results from human experts under close-book conditions would help us better understand the benchmark’s upper bound, which is highly valuable for this evaluation set.

**Questions:**

1. It is necessary to provide more details about the rule-based NLP pipeline used to construct the dataset.
2. What is the specific process for the clinical expert’s verification? Why was only one expert used, rather than multiple experts for cross-verification?
3. Most of the models evaluated in the paper are under 10 billion parameters, with only one model, Llama3-70B Instruct, having 70 billion parameters. More 70 billion parameter models should be assessed to enhance the comprehensiveness of the evaluation, such as Med42, ClinicalCamel, and others.
4. While GPT-4o performs over 70 on the L1 setting other 3 task types, its performance in procedure prediction is only 29.80, even lower than that of Llama3-70B Instruct. I believe it is essential to conduct further case studies to uncover the underlying reasons for this discrepancy.

---

> ### Comment · Reviewer_N2Z1 · 2024-12-02
> **Looking forward to the author response**
>
> This paper generally offers a valuable benchmark for evaluating LLMs’ proficiency in clinical diagnosis, and the proposed benchmark has the potential to facilitate the development of medical LLMs. If the authors can address my concerns in detail, I would be open to updating my score.

---

### Meta-Review · Area_Chair_9iQe · 2024-12-21

**Metareview:**

The paper introduces CliBench, a benchmark using MIMIC-IV to evaluate LLMs in clinical decision-making tasks, providing insights into their potential, limitations, and suitability for real-world healthcare applications.

Strengths
- CliBench introduces a benchmark for evaluating LLMs in clinical decision-making, covering diverse and realistic tasks like diagnoses, treatments, lab tests, and prescriptions
- The benchmark employs structured medical ontologies (ICD-10, LOINC, ATC) for fine-grained assessments

Weaknesses
- Tasks like ICD-10 code prediction align more with medical billing than real-world clinical decision-making, raising doubts about the benchmark's relevance for actual clinical workflows
- The reliance on zero-shot settings and lack of few-shot or prompt variation, and the lack of human clinician baselines all adds up to limited evaluation
- Potential data leakage from the MIMIC-IV dataset, reliance on a single clinical center's data, and exclusion of negative cases undermine the benchmark's validity and generalizability.

**Additional Comments On Reviewer Discussion:**

During the rebuttal, authors acknowledged core limitations of their work such as lack of prompt variation or lack of human baselines, but mostly promised to follow up in the future work.

---

### Decision · Program_Chairs · 2025-01-22

Reject